# Smartphone-Integrated User-Friendly Electrochemical Biosensor Based on Optimized Aptamer Specific to SARS-CoV-2 S1 Protein

**DOI:** 10.3390/s25216579

**Published:** 2025-10-25

**Authors:** Arzum Erdem, Huseyin Senturk, Esma Yildiz

**Affiliations:** Department of Analytical Chemistry, Faculty of Pharmacy, Ege University, 35040 İzmir, Türkiye

**Keywords:** SARS-CoV-2 S1 protein, COVID-19, optimized-aptamer, electrochemical biosensor, differential pulse voltammetry, smartphone-connected potentiostat, infectious disease

## Abstract

**Highlights:**

**What are the main findings?**
A disposable and portable voltammetric biosensor using specifically optimized aptamers (Optimers) was developed for SARS-CoV-2 S1 protein detection.The sensor achieved ultralow detection limits (18.80 ag/mL in buffer; 14.42 ag/mL in artificial saliva) with excellent selectivity against interfering proteins.

**What is the implication of the main finding?**
This study demonstrates, for the first time, the use of optimized aptamers to construct a voltammetric biosensor for highly sensitive SARS-CoV-2 detection.The integration with a smartphone-connected potentiostat highlights its potential for practical point-of-care diagnostics and outbreak management.

**Abstract:**

COVID-19, caused by SARS-CoV-2, has created unprecedented global health challenges, necessitating rapid and reliable diagnostic strategies. The spike (S) protein, particularly its S1 subunit, plays a critical role in viral entry, making it a prime biomarker for early detection. In this study, we present a disposable, low-cost, and portable electrochemical biosensor employing specifically optimized aptamers (Optimers) for SARS-CoV-2 S1 recognition. The sensing approach is based on aptamer–protein complex formation in solution, followed by immobilization onto pencil graphite electrodes (PGEs). The key parameters, including aptamer concentration, interaction time, redox probe concentration, and immobilization time, were systematically optimized by performing electrochemical measurement in redox probe solution containing ferri/ferrocyanide using differential pulse voltammetry (DPV) technique.Under optimized conditions, the biosensor achieved an ultralow detection limit of 18.80 ag/mL with a wide linear range (10^−1^–10^4^ fg/mL) in buffer. Importantly, the sensor exhibited excellent selectivity against hemagglutinin antigen and MERS-CoV-S1 protein, while maintaining high performance in artificial saliva with a detection limit of 14.42 ag/mL. Furthermore, its integration with a smartphone-connected portable potentiostat underscores strong potential for point-of-care use. To our knowledge, this is the first voltammetric biosensor utilizing optimized aptamers (Optimers) specific to SARS-CoV-2 S1 on disposable PGEs, providing a robust and field-deployable platform for early COVID-19 diagnostics.

## 1. Introduction

Electrochemical biosensors have emerged as rapidly advancing analytical platforms in numerous fields, including medicine, environmental monitoring, and food safety, primarily due to their capability of directly converting biological interactions into measurable electrical signals [1]. These sensors possess distinct advantages over traditional analytical methods, such as high signal-to-noise ratios even at small sample volumes, rapid response times, suitability for miniaturization, and low production costs. Recent advancements in surface architectures supported by nanomaterials have significantly enhanced sensor performance, enabling extremely low detection limits [2].

Aptamer-based sensing platforms make use of single-stranded DNA or RNA molecules, called aptamers, produced through SELEX, which provide exceptional specificity and strong binding interactions with target analytes [3,4]. Aptamers offer several advantages over antibodies, including ease of chemical synthesis, reproducibility, straightforward immobilization on solid supports, and structural modifiability [5]. Upon binding their targets, aptamers undergo conformational changes, which significantly influence the intensity of the electrochemical signals. This phenomenon allows selective and highly sensitive detection of target analytes through various transduction methods such as amperometric, voltammetric, and impedimetric techniques.

The outbreak of COVID-19, triggered by the SARS-CoV-2 virus, first appeared in late 2019 and rapidly developed into a global health emergency. This pathogen possesses a single-stranded RNA genome with positive polarity and gains entry into host cells via the binding of its S1 subunit, located on the spike (S) glycoprotein, to the human angiotensin-converting enzyme 2 (ACE2) receptor [6]. The virus’s high mutation rate, contagiousness, and potential for multi-organ damage underscore the necessity for early, accurate, and sensitive diagnostic methods [7,8].

Real-time RT-PCR, the clinical gold standard, is highly sensitive but limited for widespread screening due to its lengthy processing time, expensive infrastructure, and requirement for trained personnel [9]. Immunochromatographic tests targeting antigens or antibodies provide rapid results but are hindered by lower sensitivity and potential false-negative or false-positive outcomes [10]. These limitations have emphasized the importance of developing portable, user-friendly, and point-of-care electrochemical biosensors [2].

Several electrochemical aptasensors targeting COVID-19 diagnosis have been reported in the literature. Tabrizi et al. (2022) [11] developed an electrochemical impedance spectroscopy-based aptasensor by immobilizing thiolated aptamers on carbon nanofiber-gold nanoparticle (CNF–AuNP) modified screen-printed carbon electrodes to detect the receptor-binding domain (RBD) of SARS-CoV-2. The authors highlighted dense aptamer coverage facilitated by Au-S covalent bonds and emphasized that nanomaterial modification considerably improved the electrode’s effective surface area and electron transfer kinetics. The sensor achieved a detection limit of 7.0 pM over a dynamic concentration range spanning from 0.01 to 64 nM. It demonstrated strong specificity even in the presence of potential interfering biomolecules such as human serum albumin, various immunoglobulins (IgA, IgG, IgM), and influenza virus antigens, and successfully enabled the detection of SARS-CoV-2 RBD in human saliva samples. Lasserre et al. (2022) [12] adapted a low-cost, gold-coated polyester electrode derived from commercial blood glucose test strips to develop a label-free electrochemical impedance-based aptasensor. This sensor utilized a 33-nucleotide truncated aptamer immobilized via thiol-gold chemistry to specifically detect the S1 protein. The authors demonstrated clear differentiation between recombinant S1 protein and interleukin-6 negative control at 80 ng/mL, and the sensor’s successful application in clinical samples was also reported. Notably, this study represents the first report of the aptamer sequence employed in the present work, providing the foundational evidence for its binding specificity and applicability in electrochemical biosensing. In our previous study utilizing optimized aptamers, we reported an impedimetric biosensor for the determination of the S1 protein [13]. The interaction between the aptamer and its target was initially carried out in the solution phase, after which the resulting complex was immobilized onto a PGE. Electrochemical impedance spectroscopy (EIS) was employed for measurements using the [Fe(CN)_6_]^3−/4−^ redox system. The developed sensor achieved a detection limit of 8.80 ag/mL across a dynamic range spanning from 10^1^ to 10^6^ ag/mL. It demonstrated excellent selectivity over structurally related proteins such as MERS-CoV-S1 and influenza hemagglutinin, and its feasibility in artificial saliva samples was also confirmed. Adeel et al. (2022) [14] developed a label-free electrochemical aptasensor by immobilizing a thiolated DNA aptamer onto gold nanoparticle-coated flexible carbon cloth electrodes to rapidly detect SARS-CoV-2 spike protein (SP). Employing DPV and chronopotentiometry (CP), the sensor monitored signal variations of the [Fe(CN)_6_]^3−/4−^ redox probe. They reported low detection limits of 0.11 ng/mL in phosphate-buffered saline (PBS) buffer and 0.167 ng/mL in diluted human saliva. In their study, Curti et al. [15] developed an electrochemical sensor platform based on screen-printed electrodes modified with single-walled carbon nanotubes (SWCNT-SPEs), onto which a redox-tagged DNA aptamer selective for the RBD of the SARS-CoV-2 spike protein S1 subunit was immobilized. The target-induced aptamer folding reduced electron transfer efficiency between the redox label and electrode surface, resulting in suppressed amperometric signals. The sensor exhibited high specificity and an LOD of 7 nM, demonstrating successful performance in buffer solutions and artificial viral transport media commonly used for nasopharyngeal swab samples without cross-reactivity with non-target viral proteins.

In the present study, we developed an electrochemical biosensor specifically targeting the SARS-CoV-2 S1 protein. Various experimental parameters, such as aptamer–protein interaction procedure, redox probe concentration, aptamer concentration, interaction time, and immobilization time onto electrode surfaces, were systematically optimized. Under optimal conditions, the limit of detection was calculated. Selectivity was thoroughly investigated against the Hemagglutinin antigen (HA) and MERS-CoV-S1 proteins. Furthermore, the sensor’s applicability was validated using artificial saliva samples, where selectivity was also confirmed. The sensor’s potential for point-of-care applications was demonstrated by integrating the sensing platform with a smartphone-compatible portable potentiostat, achieving reliable detection of the S1 protein in both PBS and artificial saliva environments. In contrast to previous electrochemical aptasensors that primarily employed conventional aptamers and impedimetric readouts, the present study introduces, for the first time, a voltammetric biosensing platform based on specifically optimized aptamers and disposable PGEs. This unique integration enables ultrasensitive detection of the SARS-CoV-2 S1 protein at the ag/mL level and further demonstrates direct compatibility with smartphone-operated portable potentiostats. Accordingly, the proposed strategy not only offers a rapid, cost-effective, and field-deployable alternative to conventional diagnostic methods but also provides a versatile blueprint adaptable to future infectious disease outbreaks. Also, the optimized aptamer-based electrochemical platform aims to overcome limitations associated with conventional molecular and immunological methods, offering a rapid, sensitive, and cost-effective alternative for early COVID-19 diagnosis and epidemic management.

## 2. Materials and Methods

The optimized aptamer specific to the S1 protein was supplied by Aptamer Group (York, UK) [16], the company responsible for its development, optimization, and patent protection targeting the SARS-CoV-2 S1 protein. The aptamer utilized in this study was modified at its 5′ terminus with an amino group by the supplier to enable effective immobilization onto the electrode surface. The binding affinity and specificity of the aptamer toward the target analyte were originally demonstrated by Lasserre et al. [12]. In their study, the dissociation constant (*Kd*) values were reported as 10.17 ± 0.07 nM for the SARS-CoV-2 WT S1 domain, 1.19 ± 0.04 nM for the SARS-CoV-2 WT trimer, and 11.07 ± 0.10 nM for the SARS-CoV-2 B.1.617.2 S1 domain, indicating a high binding affinity and selectivity of the aptamer for different forms of the viral protein.

As instructed by the manufacturer, the aptamer was prepared in an appropriate binding buffer. This solution consists of MES monohydrate (Sigma), MgCl_2_ (Sigma), CaCl_2_ (Sigma), NaCl (Sigma), KCl (Sigma), Na_2_SO_4_ (Sigma), Tween 20 (Sigma), and BSA (Sigma). The SARS-CoV-2 S1 protein was obtained from Sino Biological. A stock solution of the protein was prepared in ultra-pure water at a concentration of 250 μg/mL and stored at −80 °C until use. Working solutions of the S1 protein were prepared by dilution in PBS (50 mM, pH 7.40).

Detailed information regarding the instruments and other reagents used in this study is provided in the Appendix A.

Experimental Procedure: An electrochemical biosensor for the S1 protein was developed using an S1 protein-specific optimized aptamer. The fabrication process consisted of the following steps:

(i) Electrochemical Activation of the PGE: The PGE was electrochemically activated by applying a potential of +1.2 V for 30 s in 500 mM acetate buffer solution (ABS, pH 4.80).

(ii) Chemical Activation via EDC/NHS Coupling: A freshly prepared solution of 5 mM *N*-(3-Dimethylaminopropyl)-*N′*-ethylcarbodiimide (EDC)/8 mM *N*-Hydroxysuccinimide (NHS) in 2-(*N*-Morpholino) ethanesulfonic acid (MES) solution (50 mM, pH 6.0) was used for surface activation. The PGE was immersed in this solution for 60 min at room temperature.

(iii) Aptamer-S1 Interaction: The aptamer was incubated with the S1 protein in the solution phase under constant stirring at 400 rpm for 15 min to allow specific binding.

(iv) Immobilization of the PGE surface: The EDC/NHS-activated PGE was immersed in the pre-formed aptamer-S1 protein complex solution for 30 min to achieve surface immobilization. A rinsing step with PBS was applied to the electrode for 5 s. This procedure ensured the removal of weakly bound species.

(v) Electrochemical Measurements: Electrochemical responses were recorded using the DPV in a 5 mM [Fe(CN)_6_]^3−/4−^ redox solution prepared in 0.1 M KCl. This technique offers advantages in terms of signal resolution and stability, reduces capacitive current contributions, and provides improved sensitivity for monitoring surface modification events arising from aptamer–protein interactions. The DPV measurements were conducted within a potential window of −0.2 V to +1.0 V, using a pulse amplitude of 50 mV and a scan rate of 50 mV/s.

Application of the Developed Biosensor in Artificial Saliva Samples: To evaluate the practical utility of the proposed biosensor for S1 protein detection within a complex biological environment, experiments were performed using artificial saliva as the test medium. The artificial saliva was formulated following the procedure described by Vozgirdaite et al. [17]. Briefly, ultrapure water was used to dissolve 4 mg/mL urea, 4 mg/mL porcine gastric mucin, 0.6 mg/mL Na_2_HPO_4_, 0.3 mg/mL NaHCO_3_, 0.6 mg/mL CaCl_2_, 0.4 mg/mL KCl, and 0.4 mg/mL NaCl. The mixture was sonicated in an ultrasonic bath for 20 min to ensure thorough dissolution and uniformity. Subsequently, the pH was adjusted to 7.2 using 0.1 M NaOH. The resulting solution was stored at 4 °C until needed. Prior to use, the artificial saliva was diluted with PBS at selected ratios.

## 3. Results and Discussion

Figure 1 illustrates the schematic design of the electrochemical biosensor constructed for the selective detection of the S1 protein.

The electrochemical and microscopic characterization studies of the biosensor developed for the S1 protein were comprehensively investigated in our previous work [13]. Therefore, in the current study, optimization experiments were first conducted for the biosensor based on the DPV technique.

To evaluate the interaction strategies in the biosensor system, two distinct procedures, solution-phase and electrode-surface interactions, were investigated. In the solution-phase interaction procedure, 0.3 nM aptamer was incubated with 1 ng/mL S1 protein under stirring for 5 min. The resulting aptamer–protein complex was then immobilized onto the electrode surface for 15 min [13]. After immobilization, DPV measurements were performed in a 2.5 mM redox solution. In the electrode-surface interaction procedure, 0.3 nM aptamer was first immobilized on the electrode surface for 30 min. Subsequently, the aptamer-modified electrodes were incubated in 1 ng/mL S1 protein solution for 30 min to allow the interaction to occur on the surface [13]. Following this interaction, DPV measurements were conducted in the same redox probe solution. In the control groups for both procedures, the same steps were followed without the use of an aptamer. As illustrated in Figure 2A,B, and detailed in Appendix A, both procedures exhibited a reduction in current following the immobilization of the S1-specific aptamer onto the electrode surface. This decline is ascribed to the electrostatic repulsion between the negatively charged phosphate backbone of the aptamer and the anionic redox couple, which impedes efficient electron transfer at the electrode–electrolyte interface [18]. Additionally, in the presence of the S1 target protein, further current suppression was observed due to steric hindrance resulting from the conformational changes upon aptamer–protein binding [18]. To verify the absence of background signals from the bare electrode (PGE), control measurements were performed in 0.1 M KCl solution. No signal was observed under these conditions, indicating that no interfering signal at around 0.2 V originated from the bare electrode and that all observed signals in the redox probe solution were attributable solely to the redox probe. A comparison of both procedures revealed that the surface-based interaction led to a 10.31% decrease in current, whereas the solution-phase interaction yielded a greater decrease of 13.12% relative to the aptamer control group (Figure 2A,B and Appendix A). Given the more pronounced and reproducible signal suppression observed with the solution-phase approach, this procedure was selected for subsequent experiments.

Following the selection of the optimal interaction procedure, the effect of redox solution concentration on the biosensor response was also examined. After aptamer–S1 protein interaction and subsequent immobilization, DPV measurements were performed using 2.5 mM and 5 mM redox solutions. As shown in Figure 2C,D and Appendix A, the highest and most reproducible current decrease (14.08% with %RSD = 4.91, *n* = 3) was obtained using the 5 mM redox solution. Therefore, all further experiments were conducted using a 5 mM redox solution.

Following the determination of the optimal redox probe concentration, the effect of chemical activation on the biosensor response was investigated. Chemical activation was performed using a covalent coupling agent pair composed of EDC and NHS. The EDC/NHS mixture was prepared in PBS and MES solutions. The PGE surface was chemically activated by immersion in the solution for 1 h. Subsequently, electrodes were immersed in the aptamer–S1 protein complex solution prepared by solution-phase interaction, allowing immobilization of the complex onto the electrode surface. DPV measurements were then carried out. As presented in Figure 2E,F and Appendix A, the most significant current decrease after aptamer–S1 protein interaction, compared to the control group, was observed when the electrodes were activated using EDC/NHS in MES buffer, with a reproducible signal suppression of 16.24% (RSD = 2.96%, *n* = 3). Based on these findings, the MES buffer was selected as the activation medium for further studies.

Following this, the effect of aptamer concentration on the biosensor response was systematically evaluated under solution-phase interaction conditions. Aptamer solutions at varying concentrations (0.003, 0.03, 0.3, 3, and 30 nM) were incubated with 1 ng/mL S1 protein. The resulting aptamer–protein complexes were immobilized onto chemically activated electrodes. DPV measurements revealed that the most pronounced and reproducible current suppression (16.24% decrease, RSD = 2.96%, *n* = 3) was obtained with 0.3 nM aptamer (Figure 2G,H and Appendix A). Thus, 0.3 nM was determined as the optimal aptamer concentration. Furthermore, control measurements were performed using the S1 protein without the aptamer under the same experimental conditions. In this control group, a significantly higher current response (320.32 ± 6.42 µA; *n* = 3) was recorded, further validating the aptamer-specific response of the biosensor.

After optimizing the aptamer concentration, the effect of interaction time between the aptamer and the S1 protein on biosensor performance was investigated. In this study, 0.3 nM aptamer was incubated with 1 ng/mL S1 protein in the solution phase for varying durations (5, 15, and 30 min). The aptamer–protein complexes were then immobilized onto EDC/NHS-activated electrodes. DPV measurements were performed in a redox solution. As shown in Figure 2I,J and Appendix A, analysis of current changes relative to the aptamer control group revealed that the most substantial and reproducible current decrease (18.10% decrease, RSD = 2.86%, *n* = 3) occurred after a 15 min interaction. Therefore, 15 min was selected as the optimal interaction time for the aptamer-S1 protein binding.

Following the determination of the optimal interaction time, the final parameter evaluated in the optimization studies was the immobilization time of the aptamer–S1 protein complex onto the electrode surface. After 15 min of solution-phase interaction between the aptamer and S1 protein, the resulting complex was immobilized onto EDC/NHS-activated electrodes for varying durations (15, 30, and 60 min). DPV measurements were subsequently performed, and the results are presented in Figure 2K,L and Appendix A. Analysis of the data revealed that the highest current decrease relative to the control group (34.37% decrease) was achieved with a 30 min immobilization period. Based on these findings, 30 min was selected as the optimal immobilization time.

Appendix A summarizes all experimentally optimized parameters pertaining to the development of the S1 protein-specific aptamer-based electrochemical biosensor.

Following the optimization of experimental parameters, the analytical performance of the developed biosensor for S1 protein was evaluated. In this context, the effect of increasing S1 protein concentrations on the biosensor response was examined over a wide dynamic range spanning from 0.1 fg/mL to 100 pg/mL (Appendix A). As the S1 protein concentration increased, a gradual decrease in current was observed. This phenomenon can be attributed to the steric hindrance caused by the increasing number of aptamer–protein complexes formed on the electrode surface. At higher analyte concentrations, more binding events occur, leading to the formation of larger molecular assemblies that physically obstruct the access of redox probe molecules to the electrode surface. This growing physical barrier results in a progressive reduction in electron transfer and thus current signal [11,18].

Through this mechanism, quantitative determination of the S1 protein was achieved. A calibration plot was generated across the concentration range of 10^−1^ to 10^4^ fg/mL, displaying a linear relationship (Figure 3A,B). The regression equation obtained from the calibration plot was: I/µA = −14.62 log (C_S1 Protein_/fg mL^−1^) + 151.77, with a correlation coefficient of R^2^ = 0.99. According to the IUPAC method [19], the LOD was calculated to be 18.80 ag/mL. Although the linear detection range (10^−1^ to 10^4^ fg/mL) obtained by DPV was slightly higher in the lower limit compared to our previous EIS-based study [13], this difference arises from the intrinsic detection principles of the two electrochemical techniques. EIS exhibits higher sensitivity toward small interfacial resistance changes, while DPV offers faster response with better reproducibility for practical applications. Furthermore, the sensitivity of the biosensor in buffer solution was determined by dividing the slope of the calibration curve (14.62) by the effective surface area of the pencil graphite electrode (0.2796 cm^2^) [13], yielding a value of 52.29 µA·mL·g^−1^·cm^−2^.

In addition, the intra-day and inter-day reproducibility of the developed biosensor was assessed in the presence of 1000 fg/mL S1 protein using two independently prepared aptasensors under identical conditions on two separate days. Intra-day reproducibility was assessed using two aptasensors that were independently prepared and tested under identical conditions on the same day (*n* = 2). Inter-day reproducibility was evaluated using two independently prepared aptasensors per day on two separate days (*n* = 4 in total). The average current responses and RSD for the first and second days were calculated as 108.40 ± 9.22 µA (RSD = 8.51%; *n* = 2) and 109.07 ± 3.35 µA (RSD = 3.07%; *n* = 2), respectively. For inter-day reproducibility, the average current and RSD values obtained from four biosensors were 108.74 ± 5.68 µA (RSD = 5.22%; *n* = 4). These results clearly demonstrate that the developed biosensor exhibits high reproducibility and delivers precise and reliable analytical performance across different fabrication batches and time points.

Various aptasensor studies reported in the literature for the electrochemical determination of SARS-CoV-2 are summarized in Table 1.

A variety of electrochemical aptasensor studies have been reported for SARS-CoV-2 detection. These studies differ in terms of electrode preparation, complexity, assay duration, detection techniques, and point-of-care (POC) applicability. Compared to the studies listed in Table 1, a lower detection limit was achieved in the present work than in other reported approaches, except for our previous study based on the EIS technique. In addition, compared to existing electrochemical aptasensor platforms, our method offers a rapid, practical, and highly sensitive strategy for SARS-CoV-2 spike protein detection. Its relatively short overall preparation time, rapid analytical measurement, solution-phase hybridization, and compatibility with portable instrumentation make it a promising candidate for real-time, point-of-care diagnostic applications. Moreover, the developed biosensor can be adapted for multiplexed analyses, allowing the simultaneous detection of multiple samples or analytes. This multiplexing capability would enable parallel measurements within a single run, thereby significantly reducing the overall analysis time and improving testing efficiency.

The selectivity of the developed S1 protein aptamer-based biosensor was evaluated in buffer medium against potential interfering agents, specifically Hemagglutinin antigen (HA) and MERS-CoV S1 (MERS) protein. Selectivity experiments were conducted at three different concentrations, 10 fg/mL (Figure 3C), 100 fg/mL (Figure 3D), and 1000 fg/mL (Figure 3E), where each agent was tested individually. Furthermore, mixtures of these agents with 100 fg/mL S1 protein were also evaluated to assess potential cross-reactivity (Figure 3F). When tested individually at 10 fg/mL, HA and MERS induced current decreases of 5.23% and 6.21%, respectively, relative to the control. In contrast, the presence of S1 protein at the same concentration resulted in a 24.63% signal decrease. At 100 fg/mL, HA and MERS caused signal reductions of 10.21% and 16.60%, while S1 protein alone yielded a 31.47% decrease. At the highest concentration (1000 fg/mL), the current decreased by 14.43% with HA and 12.12% with MERS, whereas the S1 protein caused a 41.28% signal drop. In mixed-sample experiments containing 100 fg/mL S1 protein and either HA or MERS, the biosensor responses were calculated as 99.35% and 99.06%, respectively, relative to the response obtained with 100 fg/mL S1 protein alone, which was taken as 100% (Figure 3F). To confirm the statistical differences between the S1 protein and potential interfering agents such as HA and MERS proteins, Student’s *t*-test analysis was performed. The *p*-values were calculated under the assumption that the sample groups exhibited equal variances, in order to determine whether the average current responses (*n* = 3) obtained from each sample differed significantly. Within a 95% confidence interval, the calculated *p*-values were 0.036 for 10 fg/mL HA-S1 protein, 0.006 for 10 fg/mL MERS-S1 protein, 0.003 for 100 fg/mL HA-S1 protein, 0.019 for 100 fg/mL MERS-S1 protein, 0.016 for 1000 fg/mL HA-S1 protein, and 0.001 for 1000 fg/mL MERS-S1 protein. As all *p*-values are below the statistical significance threshold of 0.05, the results confirm the presence of statistically significant differences between the compared groups. These findings indicate that, relative to other potential interferents, the aptamer exhibits a high binding affinity toward its cognate S1 protein, thereby demonstrating the high selectivity of the developed biosensor for S1 protein. For comparison, the selectivity performance of the present DPV-based biosensor was evaluated alongside our previously reported EIS-based platform detecting the S1 protein [13]. In both approaches, the biosensors exhibited excellent discrimination between the S1 protein and potential interferents (HA and MERS). In the EIS-based system, the presence of S1 protein yielded markedly higher Rct variations compared to negligible changes for HA and MERS, while the DPV-based sensor showed higher current suppression for S1 than for these interferents at corresponding concentrations. Moreover, in mixed-sample experiments, the biosensor maintained nearly identical responses to those obtained with pure S1 protein, confirming insignificant cross-reactivity. These findings demonstrate that the optimized aptamer retains strong binding specificity toward the S1 protein, irrespective of the electrochemical transduction technique employed, and highlight the robustness and versatility of the sensing platform.

To assess the real-sample applicability of the developed biosensor, experimental studies were conducted in an artificial saliva medium. As a first step, the optimal dilution ratio of artificial saliva was determined. Artificial saliva was prepared as described in the Section 2 and diluted in PBS at ratios of 1:10, 1:20, and 1:50. In each dilution condition, 10 fg/mL S1 protein was spiked into the artificial saliva, followed by incubation with the aptamer under optimized conditions. Among the tested dilutions, the most significant current decrease relative to the control group was observed at a 1:20 dilution (23.85% decrease; RSD = 5.27%, *n* = 3), indicating it as the optimal dilution (Appendix A). This finding is consistent with our previous impedance-based study [13], where the same dilution ratio was also identified as optimal.

Subsequently, S1 protein was prepared in artificial saliva at concentrations ranging from 0.1 fg/mL to 100 pg/mL to evaluate the biosensor’s analytical response (Appendix A). A calibration curve and voltammograms were constructed over the linear range of 10^−1^ to 10^4^ fg/mL (Figure 4A,B). The regression equation derived from the calibration plot was: I/µA = −8.43 log (C_S1 Protein_/fg mL^−1^) + 114.46, with a correlation coefficient of R^2^ = 0.99. According to the IUPAC method [19], the LOD was calculated to be 14.42 ag/mL. In the artificial saliva matrix, the biosensor exhibited a sensitivity of 30.15 µA·mL·fg^−1^·cm^−2^, calculated by dividing the calibration slope (8.43) by the active surface area of the PGE (0.2796 cm^2^).

Moreover, the intra-day and inter-day reproducibility of the developed biosensor was further evaluated in artificial saliva in the presence of 10 fg/mL S1 protein. For this purpose, two independently prepared aptasensors under identical conditions on two different days were tested. Intra-day reproducibility was assessed using two aptasensors that were independently prepared and tested under identical conditions on the same day (*n* = 2). Inter-day reproducibility was evaluated using two independently prepared aptasensors per day on two separate days (*n* = 4 in total). The average current responses and RSD were found to be 104.03 ± 0.94 µA (RSD = 0.90%; *n* = 2) on the first day and 109.62 ± 5.85 µA (RSD = 5.33%; *n* = 2) on the second day. In the assessment of inter-day reproducibility, the average current response and RSD calculated from four biosensors were 106.82 ± 4.70 µA (RSD = 4.40%; *n* = 4). These results clearly demonstrate that the biosensor delivers precise and reliable analytical performance even in complex biological matrices such as artificial saliva, thereby confirming its robustness and reproducibility under practical conditions. In addition, the concentration values selected for the reproducibility studies were intentionally chosen to represent different regions of the calibration curve. In PBS, reproducibility was examined at 1000 fg/mL, corresponding to the upper part of the linear range, while in diluted artificial saliva, it was tested at 10 fg/mL, a level slightly below the midpoint of the calibration curve. This approach aimed to demonstrate that the developed biosensor provides consistent and repeatable responses across both low and high analyte concentrations, even within complex biological matrices.

To evaluate the analytical performance of the developed biosensor in artificial saliva, studies were carried out to determine the relative error associated with S1 protein detection. In this context, relative error values were calculated in artificial saliva at S1P concentrations of 0.1, 1, and 100 fg/mL, and the results are presented in Appendix A. Upon examination of the results presented in Appendix A, the relative error values were found to range between 7.73% and 15.58%. Additionally, the %RSD was calculated to be below 7%. These findings indicate that the developed electrochemical biosensor enables accurate and reliable analyses in saliva samples.

Moreover, the selectivity of the fabricated biosensor in artificial saliva was assessed by testing its response to hemagglutinin (HA) and MERS proteins at concentrations of 10 fg/mL (Figure 4C), 100 fg/mL (Figure 4D), and 1000 fg/mL (Figure 4E), both individually and in combination with 100 fg/mL of S1 protein (Figure 4F). When tested alone at 10 fg/mL, HA and MERS resulted in signal decreases of 2.59% and 0.10%, respectively, compared to the control, while S1 protein at the same concentration caused a 23.85% decrease. At 100 fg/mL, HA and MERS caused signal reductions of 6.59% and 2.12%, respectively, while S1 protein yielded a 30.91% decrease. At 1000 fg/mL, HA and MERS induced decreases of 4.40% and 3.43%, respectively, while the S1 protein caused a 38.08% decrease. In the mixture studies, the biosensor responses in the presence of both S1 protein and either HA or MERS were calculated as 98.65% and 96.88%, respectively, relative to the signal obtained with 100 fg/mL S1 protein alone, which was considered as 100% (Figure 4F). To confirm the selectivity of the biosensor, a Student’s *t*-test was performed in artificial saliva, comparing the responses from the S1 protein with those obtained for possible interfering agents, including HA and MERS proteins. Assuming equal variances, statistical comparisons of the average current responses (*n* = 3) confirmed significant differences among the groups. At the 95% confidence level, the *p*-values for HA-S1 and MERS-S1 proteins at 10, 100, and 1000 fg/mL were found to be 0.005, 0.035, 0.004, 0.0002, 0.014, and 0.009, respectively. Since all *p*-values fall below the significance level of 0.05, these results confirm statistically significant differences between the tested groups. Collectively, these findings reinforce that, in comparison to other potentially interfering substances, the aptamer possesses a high binding affinity toward its specific target, the S1 protein, thereby demonstrating the excellent selectivity of the biosensor even in a complex artificial saliva matrix. Similarly, in diluted artificial saliva, the aptamer-based biosensor maintained high selectivity toward the S1 protein, exhibiting comparable discrimination to that observed in PBS. The presence of HA and MERS, either individually or in combination with S1 protein, caused only minor and statistically insignificant variations in the current response, confirming that the matrix components had a negligible influence on the biosensor’s performance. These findings are consistent with our previous EIS-based study [13], in which the same optimized aptamer demonstrated robust selectivity and negligible cross-reactivity in both buffer and artificial saliva environments. Together, these results confirm that the optimized aptamer preserves its structural stability and binding affinity under physiologically relevant conditions, ensuring consistent selectivity in complex biological media.

To demonstrate the potential of the developed biosensor for point-of-care testing, electrochemical measurements for S1 protein detection were performed using a portable potentiostat (GalvanoPlot, SolarBiotec, Izmir, Türkiye) integrated with a smartphone. The GalvanoPlot device was connected to the smartphone via a USB Type-C interface and employed a conventional three-electrode system for measurements. S1 protein solutions at various concentrations (0.1 fg/mL to 100 pg/mL) were prepared in both PBS and artificial saliva diluted at a 1:20 ratio. The same optimized procedure was followed in both media to evaluate the effect of concentration on the biosensor response (Appendix A). Calibration curves were constructed for both matrices within the linear response range of 10^−1^ to 10^4^ fg/mL (Figure 5). In PBS, the calibration equation was obtained as: I/µA = −14.42 log (C_S1 Protein_/fg mL^−1^) + 151.00 with a correlation coefficient of R^2^ = 0.99. In artificial saliva, the regression equation was: I/µA = −9.57 log (C_S1 Protein_/fg mL^−1^) + 116.56, also with R^2^ = 0.99. According to the IUPAC method [19], the LOD was calculated to be 44.24 ag/mL in PBS and 45.02 ag/mL in artificial saliva. Although the LOD values obtained using the portable potentiostat were slightly higher than those obtained with the benchtop system, this difference is mainly attributed to matrix-related effects and the higher baseline noise of the portable device. The impact was more pronounced in diluted artificial saliva due to its complex composition, which slightly reduces the signal-to-noise ratio. Nevertheless, the obtained LODs remained in the attomolar range, confirming the high sensitivity and robustness of the developed biosensor even under realistic measurement conditions. Additionally, the sensitivity of the biosensor was determined for both buffer and artificial saliva media by dividing the respective calibration slopes, 14.42 for buffer and 9.57 for artificial saliva, by the active surface area of the PGE (0.2796 cm^2^), yielding values of 51.57 and 34.23 µA·mL·fg^−1^·cm^−2^, respectively. These results clearly demonstrate that the developed biosensor, when integrated with a portable smartphone-based potentiostat, maintains its analytical performance in both standard and complex biological media, confirming its strong potential for point-of-care diagnostic applications.

Studies were conducted to calculate the relative error values of the electrochemical biosensor developed for the detection of S1 protein in artificial saliva using a smartphone-integrated portable potentiostat. In this context, relative error values were calculated in artificial saliva at S1P concentrations of 0.1, 10, and 100 fg/mL, and the obtained results are presented in Appendix A. Upon examination of the results presented in Appendix A, the relative error values were found to range between 0.00% and 18.56%. Additionally, the %RSD values were calculated to be below 5%. These findings suggest that the developed biosensor enables accurate and reliable analyses of saliva samples when used in conjunction with a smartphone-integrated portable potentiostat. Although the relative error values obtained in diluted artificial saliva slightly exceeded 15%, this deviation can be attributed to matrix-related effects such as ionic strength and viscosity differences inherent to the biological medium. In future studies, efforts will focus on reducing relative error values below 10% through the use of anti-fouling surface coatings, nanomaterial-based electrode modifications, optimization of sample preparation procedures, and advanced signal filtering in portable systems. These improvements are expected to further enhance the reproducibility and analytical accuracy of the biosensor for point-of-care diagnostic applications.

## 4. Conclusions

In the present work, an electrochemical aptasensor was designed and fabricated for the sensitive and selective detection of the S1 protein. Various experimental parameters, including the interaction procedure, redox probe concentration, aptamer concentration, aptamer–protein interaction time, and immobilization time onto the electrode surface, were systematically optimized. Under the optimized conditions, the biosensor exhibited a wide linear detection range (10^−1^–10^4^ fg/mL) in buffer medium, with a remarkably low detection limit of 18.80 ag/mL. The selectivity of the biosensor was evaluated against HA and MERS, and the results confirmed its high specificity for the S1 protein. To demonstrate the practical applicability of the biosensor, further analyses were carried out in artificial saliva as a complex biological matrix. In this medium, the biosensor retained its linear response within the same concentration range, and a detection limit of 14.42 ag/mL was achieved. Selectivity assessments in artificial saliva also confirmed that the biosensor maintained strong specificity for the S1 protein even in complex environments. To evaluate its potential for point-of-care diagnostics, the biosensor was integrated with a smartphone-connected portable potentiostat. Electrochemical measurements were successfully performed in both PBS and artificial saliva using this portable system, yielding detection limits of 44.24 ag/mL and 45.02 ag/mL, respectively, over the same linear range (10^−1^–10^4^ fg/mL). Overall, these findings demonstrate that the optimized aptamer-based biosensor is a promising tool for the rapid (seconds-scale electrochemical measurement), highly sensitive (attogram-level detection), and specific detection of SARS-CoV-2 infection. Moreover, the developed biosensor can be adapted for multiplexed analyses, allowing the simultaneous detection of multiple samples or analytes. This multiplexing capability would enable parallel measurements within a single run, thereby significantly reducing the overall analysis time and improving testing efficiency. By uniquely integrating optimized aptamers with a voltammetric sensing approach and a smartphone-compatible portable platform, this work establishes a versatile and practical framework that can be readily adapted to other clinically relevant biomarkers in future outbreaks. Future studies will also focus on validation with real clinical samples, assessment of long-term operational stability, and scalability for mass production, aiming to advance this portable sensing platform toward widespread use in point-of-care and public health applications. In addition, future improvements can be directed toward implementing redox-cycling amplification of the ferro/ferricyanide redox mediator to enhance the signal-to-background ratio and lower the detection limit, as well as adopting square-wave voltammetry (SWV) as a faster and more sensitive alternative to DPV. These strategies, as demonstrated in recent studies [25,26], may further improve analytical performance and accelerate the transition of this aptamer-based platform toward real-world clinical applications.

## Figures and Tables

**Figure 1 sensors-25-06579-f001:**
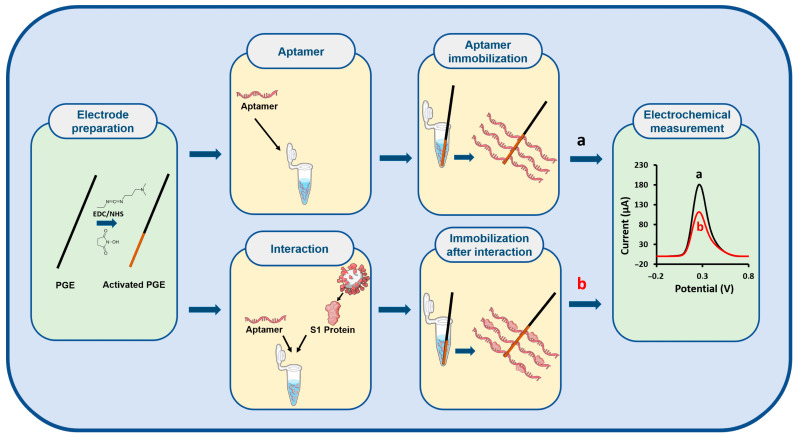
Schematic illustration of the electrochemical biosensor constructed for the specific detection of the S1 protein. The part indicated as (a) represents the procedure corresponding to the control group without the presence of the S1 protein, while the part indicated as (b) represents the procedure corresponding to the group containing the S1 protein.

**Figure 2 sensors-25-06579-f002:**
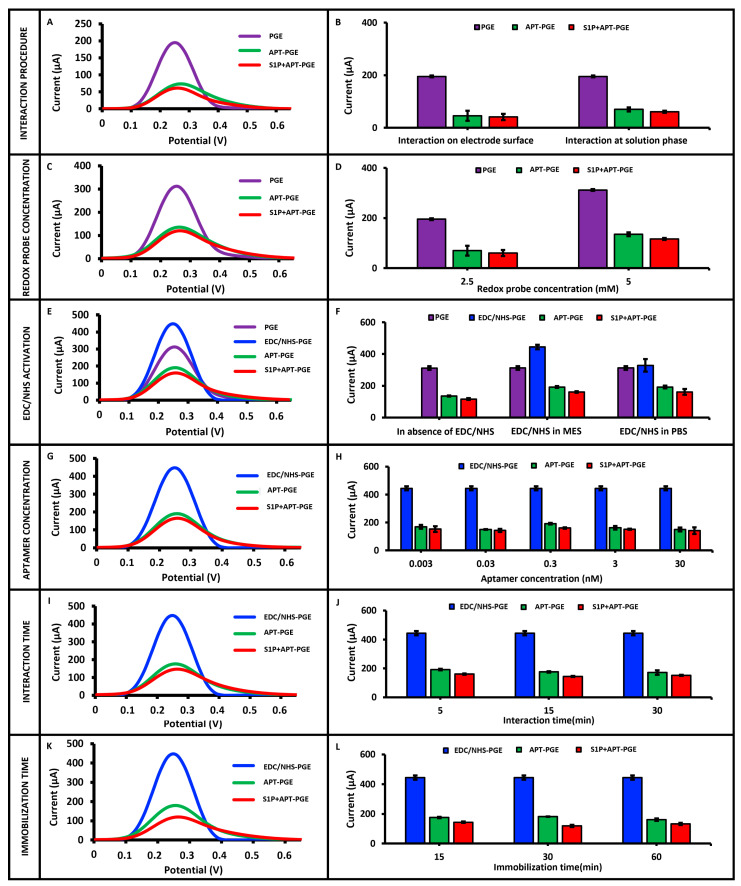
(**A**,**C**,**E**,**G**,**I**,**K**) represent DPV voltammograms obtained in 5 mM redox probe solution for the investigation of optimization parameters, including the interaction procedure, redox probe concentration, effect of EDC/NHS, aptamer concentration, interaction time, and immobilization time, respectively. (**B**,**D**,**F**,**H**,**J**,**L**) show the corresponding histograms of average current values (*n* = 3) related to each parameter.

**Figure 3 sensors-25-06579-f003:**
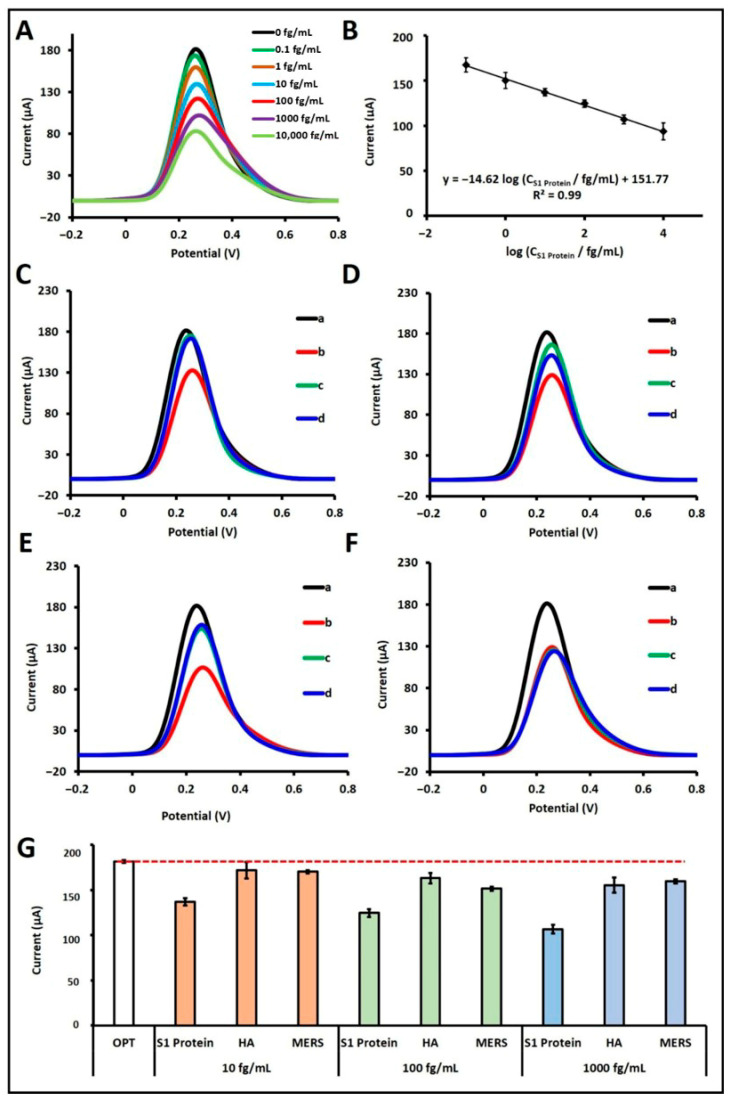
(**A**) DPV voltammograms and (**B**) corresponding calibration curve showing the average current values (*n* = 3) obtained after the interaction of aptamer with S1 protein over the concentration range of 0–10,000 fg/mL in buffer medium, measured in 5 mM redox solution. (**C**–**E**) DPV voltammograms obtained from selectivity studies conducted at 10 fg/mL, 100 fg/mL, and 1000 fg/mL concentrations, respectively. In Figures (**C**–**E**), (a) represents the aptamer control, (b) the S1 protein, (c) HA, and (d) MERS. (**F**) DPV voltammograms showing selectivity results in the mixed system at 100 fg/mL concentration, where (a) represents the aptamer control, (b) S1 protein, (c) S1 protein + HA, and (d) S1 protein + MERS. (**G**) Histograms presenting the average current values (n = 3) obtained from selectivity studies conducted against HA and MERS at 10, 100, and 1000 fg/mL.

**Figure 4 sensors-25-06579-f004:**
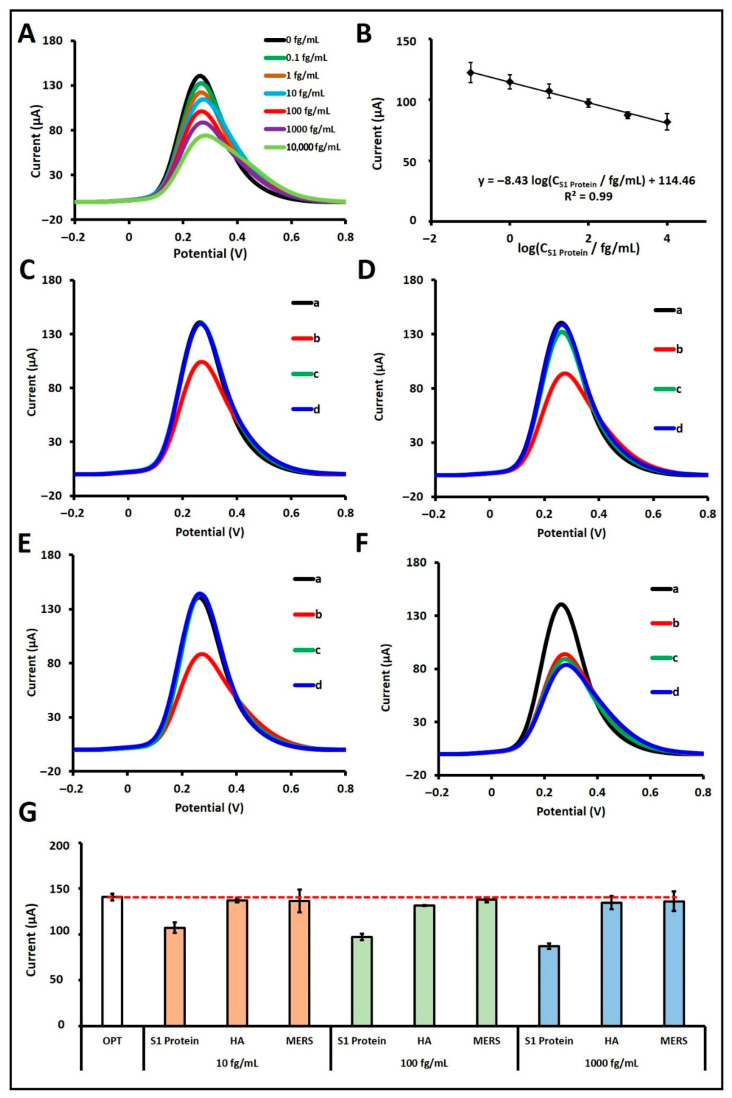
(**A**) DPV voltammograms and (**B**) corresponding calibration curve showing the average current values (*n* = 3) obtained after the interaction of aptamer with S1 protein over the concentration range of 0–10,000 fg/mL in artificial saliva medium, measured in 5 mM redox solution. (**C**–**E**) DPV voltammograms obtained from selectivity studies conducted at 10 fg/mL, 100 fg/mL, and 1000 fg/mL concentrations, respectively. In Figures (**C**–**E**), (a) represents the aptamer control, (b) the S1 protein, (c) HA, and (d) MERS. (**F**) DPV voltammograms showing the selectivity results in the mixed system at a concentration of 100 fg/mL, where (a) represents the aptamer control, (b) the S1 protein, (c) the S1 protein + HA, and (d) the S1 protein + MERS. (**G**) Histograms presenting the average current values (n = 3) obtained from selectivity studies conducted against HA and MERS at concentrations of 10, 100, and 1000 fg/mL.

**Figure 5 sensors-25-06579-f005:**
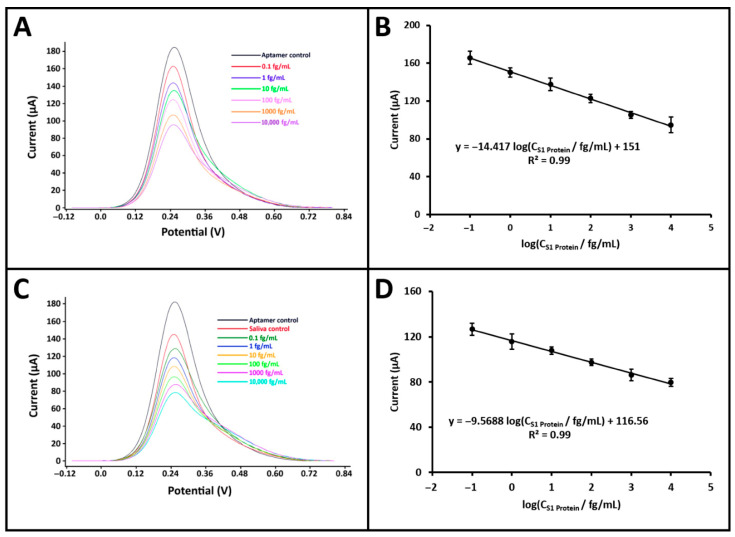
Studies on the determination of S1 protein using a smartphone-integrated potentiostat. (**A**,**B**), respectively, show the voltammograms and the corresponding calibration curve (*n* = 3) obtained after DPV measurements in a buffer medium within the concentration range of 0.1 fg/mL to 10,000 fg/mL. (**C**,**D**), respectively, show the voltammograms and the corresponding calibration curve (*n* = 3) obtained after DPV measurements in a 1:20 diluted artificial saliva medium within the concentration range of 0.1 fg/mL to 10,000 fg/mL.

**Table 1 sensors-25-06579-t001:** Various aptasensor studies reported in the literature for the electrochemical determination of SARS-CoV-2.

Analyte	Electrode	Biosensing Surface	Method	Linear Range	Detection Limit	Application	Ref.
SARS-CoV-2-RBD	Carbon-based screen-printed electrode (CSPE)	CNF–AuNP nanocomposite (CSPE/CNF–AuNP) with thiol-terminal aptamer	EIS	0.01–64 nM	7.0 pM	Human saliva samples	[11]
SARS-CoV-2 Spike S1	Thin-film gold electrodes	Thiol-terminal aptamer	EIS	-	-	Clinical patient samples	[12]
SARS-CoV-2 S1 protein	Pencil graphite electrode (PGE)	-NH_2_ linked optimized aptamer	EIS	10^1^–10^6^ ag/mL	8.80 ag/mL	Artificial saliva	[13]
SARS-CoV-2 Spike Protein	Flexible carbon cloth electrode	Gold nanoparticles with thiol functionalized DNA aptamer	DPV and CP	0–1000 ng/mL for DPV and CP	0.11 ng/mL for DPV and 37.8 ng/mL for CP	Human saliva	[14]
SARS-CoV-2 RBD	Screen-printed electrode (SPE)	Single-walled carbon nanotube with redox-tagged DNA aptamer	Amperometry	-	7 nM	Artificial viral transport media for nasopharyngeal swabs	[15]
SARS-CoV-2 RBD	Screen-printed carbon electrodes (SPCEs)	Gold nanoparticles (AuNPs) with thiol tagged DNA aptamer	EIS	10 pM–25 nM	1.30 pM (66 pg/mL)	SARS-CoV-2 pseudovirus	[20]
SARS-CoV-2 Spike RBD	Interdigitated gold electrode (IDE)	Thiolated aptamer	EIS	0.2 to 0.8 pg/mL	0.4 pg/mL	Human Nasal Fluid	[21]
RBD Protein S SARS-CoV-2	Screen-printed carbon electrode (SPCE)	Gold nanoparticles (AuNPs) with biotinylated aptamer	DPV	10–50 ng/mL	2.63 ng/mL	Saliva samples	[22]
SARS-CoV-2 spike (S) protein	Gold wire electrode	Self-assembled monolayer with aptamer	SWV	10 pM to 100 nM	-	Artificial saliva and fetal bovine serum	[23]
SARS-CoV-2 nucleocapsid protein	Gold electrode	Metal–organic frameworks MIL-53(Al) decorated with Au@Pt nanoparticles combined with thiol-modified aptamers	DPV	0.025 ng/mL–50 ng/mL	8.33 pg/mL	Serum	[24]
SARS-CoV-2 S1 protein	Pencil graphite electrode (PGE)	-NH_2_ linked optimized aptamer	DPV	10^−1^–10^4^ fg/mL	18.80 ag/mL	Artificial saliva	This study

## Data Availability

Data will be made available on request.

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
