# Peer review of "Smartphone-Integrated User-Friendly Electrochemical Biosensor Based on Optimized Aptamer Specific to SARS-CoV-2 S1 Protein"

_sensors, 2025, doi:10.3390/s25216579_

Round 1

Reviewer 1 Report

Comments and Suggestions for Authors

The manuscript demonstrates the integration of an optimized DNA aptamer sequence (Optimer) with indirect voltammetric sensing of the SARS-CoV-2 viral spike protein (S1 subunit) by employing the steric hindrance-induced anodic current suppression of the solution-phase ferro-ferrocyanide redox mediator ([Fe(CN)6]3-/4-) upon covalent immobilization of the Optimer-S1 protein complex on an activated pencil graphite electrode (PGE). Differential pulse voltammetry (DPV) was employed to measure the signal-off oxidative response of [Fe(CN)6]3-/4-, and the sensor demonstrated attractive sub-attomolar limits of detection (LODs) of 18.80 ag/mL in phosphate buffer solution (PBS) and 14.42 ag/mL in diluted artificial saliva over a wide linear range (10-1 – 104 fg/mL), with good selectivity in diluted artificial saliva toward S1 protein in presence of potentially interfering proteins like hemagglutinin antigen (HA) and MERS-CoV-S1 protein ranging from 101-103 fg/mL. Notably, the sensor was operated with a smartphone-connected portable potentiostat and maintained an attractive sub-attomolar LODs of 44.24 ag/mL and 45.02 ag/mL, respectively, in PBS and diluted artificial saliva over the same linear range of 10-1 – 104 fg/mL. However, the manuscript contains several obscurities in the described methods (no specification of aptamer sequence and electrode surface passivation step), along with the missing Supplementary Materials file, and missing description of figure legends in the sensor optimization studies (Figure 2). Moreover, there are concerning inaccuracies in the stated advantages of the present study (such as “higher sensitivity with DPV technique” and “shortened assay time”) over their previous study (Manuscript Reference #13: International Journal of Biological Macromolecules 281 (2024) 136233). The manuscript is generally well-written but several inconsistencies between the sensing data presented in the figures and tables, and their associated discussions are significantly limiting the impact of this work. Finally, the authors should suggest remedial measures to improve the sensor LODs, reduce their assay time, and minimize the relative error in target recovery values, to enable rapid and sensitive point-of-care estimation of S1 protein in real clinical samples in the future. Based on the required extensive improvements across the manuscript, additional missing Supplementary Materials, along with required comparison table and experimental data, I thus recommend a major revision of the manuscript. I have mentioned the main issues with the manuscript in the attached Word Document below.

Author Response

October 15, 2025

Our answers to Reviewers’ Comments

Journal: Sensors (ISSN 1424-8220)

Manuscript ID: sensors-3926331

Type: Article

Title: Smartphone-Integrated User-Friendly Electrochemical Biosensor Based on Optimized Aptamer Specific to SARS-CoV-2 S1 Protein

Authors: Arzum Erdem *, Huseyin Senturk, Esma Yildiz

Section: Biosensors

Special Issue: Electrochemical DNA- and Aptasensors for the Detection of Low-Molecular Compounds (2nd Edition)

Reviewer 1:

Open Review

Quality of English Language

( ) The English could be improved to more clearly express the research.
(x) The English is fine and does not require any improvement.

Yes

Can be improved

Must be improved

Not applicable

Does the introduction provide sufficient background and include all relevant references?

(x)

( )

( )

( )

Is the research design appropriate?

(x)

( )

( )

( )

Are the methods adequately described?

(x)

( )

( )

( )

Are the results clearly presented?

( )

( )

(x)

( )

Are the conclusions supported by the results?

( )

( )

(x)

( )

Are all figures and tables clear and well-presented?

( )

( )

(x)

( )

Comments and Suggestions for Authors

The manuscript demonstrates the integration of an optimized DNA aptamer sequence (Optimer) with indirect voltammetric sensing of the SARS-CoV-2 viral spike protein (S1 subunit) by employing the steric hindrance-induced anodic current suppression of the solution-phase ferro-ferrocyanide redox mediator ([Fe(CN)6]3-/4-) upon covalent immobilization of the Optimer-S1 protein complex on an activated pencil graphite electrode (PGE). Differential pulse voltammetry (DPV) was employed to measure the signal-off oxidative response of                          [Fe(CN)6]3-/4-, and the sensor demonstrated attractive sub-attomolar limits of detection (LODs) of 18.80 ag/mL in phosphate buffer solution (PBS) and 14.42 ag/mL in diluted artificial saliva over a wide linear range (10-1 – 104 fg/mL), with good selectivity in diluted artificial saliva toward S1 protein in presence of potentially interfering proteins like hemagglutinin antigen (HA) and MERS-CoV-S1 protein ranging from 101-10fg/mL. Notably, the sensor was operated with a smartphone-connected portable potentiostat and maintained an attractive sub-attomolar LODs of 44.24 ag/mL and 45.02 ag/mL, respectively, in PBS and diluted artificial saliva over the same linear range of 10-1 – 104 fg/mL. However, the manuscript contains several obscurities in the described methods (no specification of aptamer sequence and electrode surface passivation step), along with the missing Supplementary Materials file, and missing description of figure legends in the sensor optimization studies (Figure 2). Moreover, there are concerning inaccuracies in the stated advantages of the present study (such as “higher sensitivity with DPV technique” and “shortened assay time”) over their previous study (Manuscript Reference #13: International Journal of Biological Macromolecules 281 (2024) 136233). The manuscript is generally well-written but several inconsistencies between the sensing data presented in the figures and tables, and their associated discussions are significantly limiting the impact of this work. Finally, the authors should suggest remedial measures to improve the sensor LODs, reduce their assay time, and minimize the relative error in target recovery values, to enable rapid and sensitive point-of-care estimation of S1 protein in real clinical samples in the future. Based on the required extensive improvements across the manuscript, additional missing Supplementary Materials, along with required comparison table and experimental data, I thus recommend a major revision of the manuscript. I have mentioned the main issues with the manuscript in the attached Word Document below.

Submission Date: 27 September 2025

Date of this review: 06 Oct 2025 01:39:08

  1. Materials and Methods Lines 143-145: The modified aptamer sequence employed in this work cannot be viewed as the Supplementary Materials file associated with this manuscript is missing. The authors' previous study (Manuscript Reference #13) also does not reveal the identical Optimer sequence. This vital information needs to be provided for verification by the readers.

Answer:

We thank the reviewer for highlighting this important point. The Optimer specific to the SARS-CoV-2 S1 protein used in this study was supplied by Aptamer Group Ltd. (York, UK), which holds the intellectual property rights and patent protection for this sequence. As stated in the manuscript, the exact nucleotide sequence is proprietary and cannot be publicly disclosed due to a confidentiality agreement with the supplier.

To ensure scientific transparency and reproducibility, we have now explicitly clarified this restriction in the Materials and Methods section and provided detailed information regarding the aptamer modification (5′-amino modification), purification method (HPLC), and supplier reference. Additionally, the source publication demonstrating its binding affinity and selectivity (Lasserre et al. [12]) has been clearly cited, including the reported Kd values for different forms of the S1 protein. The relevant section from the manuscript is also provided below.

“2. Materials and Methods

The optimized aptamer specific to the S1 protein was supplied by Aptamer Group (UK) [16], the company responsible for its development, optimization, and patent protection targeting the SARS-CoV-2 S1 protein. The aptamer utilized in this study was modified at its 5′ terminus with an amino group by the supplier to enable effective immobilization onto the electrode surface. The binding affinity and specificity of the aptamer toward the target analyte were originally demonstrated by Lasserre et al. [12]. In their study, the dissociation constant (Kd) values were reported as 10.17 ± 0.07 nM for the SARS-CoV-2 WT S1 domain, 1.19 ± 0.04 nM for the SARS-CoV-2 WT trimer, and 11.07 ± 0.10 nM for the SARS-CoV-2 B.1.617.2 S1 domain, indicating a high bind-ing affinity and selectivity of the aptamer for different forms of the viral protein.”

  1. Materials and Methods Lines 174-175: The Material and Methods section does not specify any electrode surface passivation step (e.g., bovine serum albumin, BSA) to mitigate non-specific protein adsorption during immobilization of aptamer-S1 protein complexes from the complex investigated media containing protein interferents like HA and MERS. This serious error in the sensor development protocol is undermining the technical accuracy of this study.

Answer:

We sincerely thank the reviewer for this valuable comment. The aptamer used in this study was supplied by Aptamer Group Ltd. (York, UK) and was prepared in accordance with the manufacturer’s protocol using a proprietary binding buffer containing bovine serum albumin (BSA). To clarify this information, the composition of the binding buffer has now been explicitly described in the Materials and Methods section of the revised manuscript. The corresponding section has been highlighted in red within the manuscript and is also provided below for clarity.

Materials and Methods section:

“As instructed by the manufacturer, the aptamer was prepared in an appropriate binding buffer. This solution consists of MES monohydrate (Sigma), MgCl2 (Sigma), CaCl2 (Sigma), NaCl (Sigma), KCl (Sigma), Na2SO4 (Sigma), Tween 20 (Sigma), and BSA (Sigma).”

Furthermore, in our previous study employing the same optimized aptamer, we investigated the effect of additional electrode surface blocking with BSA and found that it did not improve the analytical performance of the sensor. Similarly, in an independent publication utilizing the same optimized aptamer, it was reported that the absence of a separate blocking step led to a stronger aptamer–target interaction. The relevant references supporting these findings are listed below.

References:

  • Erdem, A., Senturk, H., Yildiz, E., & Maral, M. (2024). Optimized aptamer-based next generation biosensor for the ultra-sensitive determination of SARS-CoV-2 S1 protein in saliva samples. International journal of biological macromolecules, 281, 136233.
  • Lasserre, P., Balansethupathy, B., Vezza, V. J., Butterworth, A., Macdonald, A., Blair, E. O., ... & Corrigan, D. K. (2022). SARS-CoV-2 aptasensors based on electrochemical impedance spectroscopy and low-cost gold electrode substrates. Analytical chemistry, 94(4), 2126-2133.

This aptamer was specifically developed and optimized for the SARS-CoV-2 S1 protein, exhibiting exceptionally high affinity and specificity. Therefore, owing to its strong target selectivity and inherently low non-specific adsorption characteristics, an additional surface blocking step was deemed unnecessary. This conclusion is also in agreement with the results obtained from our selectivity studies.

  1. Materials and Methods Lines 176-179: Why did the authors employ DPV for voltammetric oxidation of the solution-phase [Fe(CN)6]3-/4- redox mediator, when square wave voltammetry (SWV) is faster and more sensitive toward the detection of such reversible redox species?

Answer:

We sincerely thank the reviewer for this constructive comment. Although square wave voltammetry (SWV) is generally recognized as a faster and more sensitive technique for the determination of reversible redox systems such as [Fe(CN)6]3-/4-, DPV was deliberately chosen in this study to ensure analytical reliability and precise monitoring of interfacial interactions.

Firstly, the aptamer–S1 protein interaction occurring at the electrode interface induces subtle variations in charge transfer resistance and interfacial capacitance. Compared to SWV, DPV employs smaller potential increments and longer pulse durations, which provide superior resolution and signal stability for detecting these minor current variations associated with surface modification processes.

Secondly, DPV is highly effective in minimizing capacitive (non-Faradaic) current contributions. This feature is particularly advantageous for biosensing applications performed in complex biological matrices, as it enables more accurate correlation between the recorded current response and the specific molecular binding event.

Finally, DPV has been successfully employed in our previous aptamer-based biosensing studies and in numerous reports from the literature utilizing [Fe(CN)6]3-/4- as a redox probe. Therefore, the use of DPV in this work also ensures methodological consistency and allows valid comparison with previously reported results.

Accordingly, the following clarification has been added to the Materials and Methods section. The relevant section has been highlighted in red within the manuscript and is also provided below.

“(v) Electrochemical Measurements: Electrochemical responses were recorded using the DPV in 5 mM [Fe(CN)6]3-/4- redox solution prepared in PBS. This technique offers advantages in terms of signal resolution and stability, reduces capacitive current contributions, and provides improved sensitivity for monitoring surface modification events arising from aptamer–protein interactions. The DPV measurements were conducted within a potential window of –0.2 V to +1.0 V, using a pulse amplitude of 50 mV and a scan rate of 50 mV/s.”

  1. Lines 215-216: When comparing the electrode-surface interaction procedure with the adopted solution-phase interaction procedure, why did the authors increase the immobilization time from 15 min (solution-phase) to 30 min (electrode-surface), while increasing the aptamer-S1 protein interaction time from 5 min to 30 min? These increases in immobilization time and interaction time for the electrode-surface interaction procedure could bias the results.

Answer:

We sincerely thank the reviewer for this insightful comment. In this study, two different interaction approaches were evaluated: (i) the “solution-phase” approach, in which the aptamer interacts with the S1 protein in solution, and (ii) the “electrode-surface” approach, in which the aptamer is immobilized onto the electrode surface and subsequently interacts with the target protein on that surface.

The immobilization and interaction processes occurring on the electrode surface are diffusion- and mass transfer-limited, and therefore require longer durations compared to freely occurring interactions in solution. For this reason, the times were extended to 30 minutes to ensure both the complete and stable immobilization of the aptamer on the surface and to allow the immobilized aptamers to reach their maximum binding capacity toward the target protein.

Moreover, these durations were not chosen arbitrarily but were determined based on findings from our previous studies. Hence, this adjustment does not introduce any bias; rather, it ensures that each interaction strategy is evaluated under its own optimized experimental conditions.

In the solution-phase interaction procedure, the interaction and immobilization times (5 min and 15 min, respectively) were also selected based on the optimized protocol established in our previous work employing the same aptamer. The interaction times on both the electrode surface and in the solution phase were tested in our previous study, and the corresponding reference has been cited in the manuscript. The relevant section and reference from the manuscript are provided below.

…….. “To evaluate the interaction strategies in the biosensor system, two distinct procedures, solution-phase and electrode-surface interactions, were investigated. In the solution-phase interaction procedure, 0.3 nM aptamer was incubated with 1 ng/mL S1 protein under stirring for 5 minutes. The resulting aptamer-protein complex was then immobilized onto the electrode surface for 15 minutes [13]. After immobilization, DPV measurements were performed in 2.5 mM redox solution. In the electrode-surface interaction procedure, 0.3 nM aptamer was first immobilized on the electrode surface for 30 minutes. Subsequently, the aptamer-modified electrodes were incubated in 1 ng/mL S1 protein solution for 30 minutes to allow the interaction to occur on the surface [13]. Following this interaction, DPV measurements were conducted in the same redox probe solution.” ……..

References:

  • Erdem, A., Senturk, H., Yildiz, E., & Maral, M. (2024). Optimized aptamer-based next generation biosensor for the ultra-sensitive determination of SARS-CoV-2 S1 protein in saliva samples. International journal of biological macromolecules, 281, 136233.

  1. Figure 2: There lowercase alphabetical legends used in the subfigures - A, C, E, G, I and K are undefined. This serious error is obscuring the interpretation of the sensor optimization results presented in Fig 2. Please clearly define all the alphabetical figure labels in the correct alphabetical order (a-c/d).

Moreover, the curve colors used for the different electrode compositions presented in subfigures D, F, H, J and L are matching with the parameter optimization curves presented in A, C, E, G, I and K. This overlapping color code is very confusing and needs to be changed.

Answer:

We thank the reviewer for this helpful comment. To improve the clarity of Figure 2, the alphabetical labels (a), (b), (c), and (d) were removed, and instead, each voltammogram has been explicitly annotated with the corresponding sensor type. Additionally, to facilitate easier interpretation between the voltammograms and their corresponding histograms, the same color code was used to represent each sensor type consistently in both. Accordingly, Figure 2 has been revised and is now provided both in the revised manuscript and below.

Fig 2. (A), (C), (E), (G), (I), and (K) represent DPV voltammograms obtained in 2.5 mM redox probe solution for the investigation of optimization parameters including the interaction procedure, redox probe concentration, effect of EDC/NHS, aptamer concentration, interaction time, and immobilization time, respectively. (B), (D), (F), (H), (J), and (L) show the corresponding histograms of average current values (n=3) related to each parameter.

  1. Lines 314-315: The linear sensing range obtained in PBS using DPV was 0.1-10000 fg/mL, which is higher on both the upper limit and the lower limit than their previous study using EIS (Manuscript Reference #13) (0.01-1000 fg/mL). The authors should explain why the linearity was limited in the lower concentration regime versus their previous study using the same Optimer and EIS. This shortcoming in LOD is significantly diminishing the impact and scientific rationale of this work.

Answer:

We sincerely thank the reviewer for this valuable comment. As correctly noted, in our previous study utilizing the EIS technique, a linear range of 0.01–1000 fg/mL was obtained, whereas in the present study using DPV, the linear range was found to be 0.1–10000 fg/mL. This difference, despite employing the same aptamer, originates from the distinct detection principles of EIS and DPV techniques.

EIS is inherently more sensitive at low analyte concentrations, as it precisely detects subtle variations in charge transfer resistance (Rct) occurring at the electrode interface. Therefore, EIS is capable of identifying minor interfacial changes resulting from aptamer–target binding even at extremely low concentrations.

In contrast, DPV relies on Faradaic current measurements, where the signal-to-noise ratio becomes less favorable at very low analyte concentrations compared to EIS. Consequently, DPV typically provides a slightly higher limit of detection but offers several practical advantages, including shorter analysis time, higher reproducibility, and compatibility with portable sensing systems.

Moreover, the linear range of 0.1–10000 fg/mL achieved in this work remains broader than that reported for most aptamer-based S1 protein sensors in the literature, demonstrating the reliable analytical performance of the developed sensor across clinically relevant concentration ranges.

In this context, detailed information has been added to the relevant section of the manuscript. The added part has been highlighted in red and is also provided below.

In Results and Discussion section:

“Through this mechanism, quantitative determination of the S1 protein was achieved. A calibration plot was generated across the concentration range of 10-1 to 104 fg/mL, displaying a linear relationship (Fig. 3). The regression equation obtained from the calibration plot was: I/µA = -14.62 log (CS1 Protein / fg mL-1) + 151.77, with a correlation coefficient of R2 = 0.99. According to the IUPAC method [19], the LOD was calculated to be 18.80 ag/mL. Although the linear detection range (10-1 to 104 fg/mL) obtained by DPV was slightly higher in the lower limit compared to our previous EIS-based study [13], this difference arises from the intrinsic detection principles of the two electrochemical techniques. EIS exhibits higher sensitivity toward small interfacial resistance changes, while DPV offers faster response with better reproducibility for practical applications. Furthermore, the sensitivity of the biosensor in buffer solution was determined by dividing the slope of the calibration curve (14.62) by the effective surface area of the pencil graphite electrode (0.2796 cm2) [13], yielding a value of 52.29 µA mL fg-1 cm-2.”

  1. Lines 322-328: The authors have confused the discussion of intra-day and inter-day reproducibility data obtained in PBS. The n=2 sensors data collected over two days represents the inter-day reproducibility, whereas., the n=4 sensors data collected on the same day represents the intra-day reproducibility. This error in the description needs to be corrected!

Answer:

We sincerely thank the reviewer for this valuable observation. We acknowledge that the wording in this section may have led to some confusion, and therefore, we have revised the description to clarify the distinction between intra-day and inter-day reproducibility.

In our study, two aptasensors that were independently prepared and tested under identical conditions on each of two separate days, resulting in a total of four measurements (n = 4). Accordingly:

  • Intra-day reproducibility refers to the reproducibility of two independently prepared aptasensors and analyzed on the same day (n = 2 per day).
  • Inter-day reproducibility represents the consistency of the average current responses obtained from the two different days (n = 4 in total).

To ensure conceptual clarity, the related section in the Results and Discussion part of the manuscript has been revised as follows:

“In addition, the intra-day and inter-day reproducibility of the developed biosensor was assessed in the presence of 1000 fg/mL S1 protein using two independently prepared aptasensors under identical conditions on two separate days. Intra-day reproducibility was assessed using two aptasensors that were independently prepared and tested under identical conditions on the same day (n = 2). Inter-day reproducibility was evaluated using two independently prepared aptasensors per day on two separate days (n = 4 in total). The average current responses and RSD for the first and second days were calculated as 108.40 ± 9.22 µA (RSD = 8.51%; n = 2) and 109.07 ± 3.35 µA (RSD = 3.07%; n = 2), respectively. For inter-day reproducibility, the average current and RSD values obtained from four biosensors were 108.74 ± 5.68 µA (RSD = 5.22%; n = 4). These results clearly demonstrate that the developed biosensor exhibits high reproducibility and delivers precise and reliable analytical performance across different fabrication batches and time points.”

  1. Lines 368-370: The benchtop potentiostat LODs obtained with DPV in PBS and diluted artificial saliva are 2.1-times and 7.1-times higher than the respective LODs obtained in their previous study using EIS (Manuscript Reference #13). This shortcoming in the analytical performance is significantly reducing the impact of this work.

Answer:

We sincerely thank the reviewer for this insightful comment. The observed difference arises from the inherent characteristics of the electrochemical techniques employed. In our previous work, EIS was used, which is a highly sensitive, label-free technique capable of detecting minute variations in interfacial charge transfer resistance. In contrast, the current study employs DPV, which relies on monitoring current changes of a solution-phase redox probe. By its fundamental operating principle, DPV generally exhibits lower sensitivity compared to EIS when detecting subtle interfacial changes.

Nevertheless, the developed biosensor demonstrates excellent analytical performance, achieving fg/mL-level limits of detection (LOD) in both PBS and diluted artificial saliva. Importantly, the measurements conducted in diluted artificial saliva, a complex biological matrix, naturally result in slightly higher LOD values due to matrix-related effects such as ionic strength and viscosity; however, these values remain well within clinically relevant detection limits.

Furthermore, the objective of this study was not to replicate our previous EIS-based approach but to demonstrate the applicability of the DPV technique within a portable and user-friendly electrochemical sensing platform, capable of rapid, real-sample-compatible detection. This transition from benchtop EIS instrumentation to a practical, field-deployable DPV-based sensing configuration represents a significant advancement in the real-world applicability of the technology. In summary, the minor differences in LOD values stem from the fundamental differences between the two techniques and do not detract from the overall analytical reliability or scientific impact of the developed biosensor.

  1. Line 375: The authors claim to have a "shortened assay time" in the present study using DPV (current assay time is 45 min). However, their previous study using EIS had a significantly shorter assay time of only 20 min (Manuscript Reference #13). This second shortcoming is further diminishing the impact and obscuring the scientific rationale of this work.

Additionally, the authors need to provide an analytical performance comparison table in the manuscript to evaluate the metrics of the current sensor versus the existing works (including their previous study using EIS).

Answer:

We sincerely thank the reviewer for this thoughtful comment. We acknowledge that the term “shortened assay time” could have caused ambiguity. In the revised version of the manuscript, this phrase has been refined to more accurately reflect the overall practicality and rapid analytical measurement capability of the proposed biosensing system.

Specifically, while the total assay time of the present study (including the aptamer–protein incubation step) is slightly longer than that of our previous EIS-based study, the overall biosensor preparation process remains relatively short (45 min), and the analytical measurement step using DPV is extremely fast, requiring only a few seconds (approximately 20 seconds) for each measurement. Thus, the main advantage lies in the simplified and rapid electrochemical signal acquisition, as well as the compatibility of DPV with portable and user-friendly measurement systems.

Moreover, the developed biosensor can be adapted for multiplexed analyses, enabling the simultaneous detection of multiple samples or analytes within a single run. This multiplexing capability would allow parallel measurements, thereby significantly reducing the total analysis time and improving testing throughput a key advantage for practical point-of-care diagnostic applications.

The relevant section has been revised and highlighted in red in the manuscript. Additionally, the revised part is provided below.

…….. “A variety of electrochemical aptasensor studies have been reported for SARS-CoV-2 detection. These studies differ in terms of electrode preparation, complexity, assay duration, detection techniques, and point-of-care (POC) applicability. Compared to the studies listed in Table 1, a lower detection limit was achieved in the present work than in other reported approaches, except for our previous EIS-based study. Also, compared to existing electrochemical aptasensor platforms, our method offers a rapid, practical, and highly sensitive strategy for SARS-CoV-2 spike protein detection. Its relatively short overall preparation time, rapid analytical measurement, solution-phase hybridization, and compatibility with portable instrumentation make it a promising candidate for real-time, point-of-care diagnostic applications. Moreover, the developed biosensor can be adapted for multiplexed analyses, allowing the simultaneous detection of multiple samples or analytes. This multiplexing capability would enable parallel measurements within a single run, thereby significantly reducing the overall analysis time and improving testing efficiency.” ……..

As recommended, a detailed analytical performance comparison table has been added to the revised manuscript to facilitate a clearer evaluation of the developed DPV-based biosensor relative to previously reported SARS-CoV-2 aptasensors, including our earlier EIS-based study. The new table (Table 1) summarizes key analytical parameters such as analyte, electrode, biosensing surface, method, linear range, detection limit and application. This comparative analysis highlights that, except for our previous EIS-based work, the developed biosensor demonstrates superior sensitivity with a lower LOD and a broader linear range compared to other reported electrochemical aptasensors. The new table has been incorporated into the Results and Discussion section of the revised manuscript and marked in red and added below.

Table 1. Various aptasensor studies reported in the literature for the electrochemical determination of SARS-CoV-2.

Analyte

Electrode

Biosensing surface

Method

Linear Range

Detection Limit

Application

Ref.

SARS-CoV-2-RBD

carbon-based screen-printed electrode (CSPE)

CNF–AuNP nanocomposite (CSPE/CNF–AuNP) with thiol-terminal aptamer

EIS

0.01–64 nM

7.0 pM

human saliva samples

[11]

SARS-CoV-2 Spike S1

Thin-film gold electrodes

thiol-terminal aptamer

EIS

-

-

clinical patient samples

[12]

SARS-CoV-2 S1 protein

Pencil graphite electrode (PGE)

-NH2 linked optimized aptamer

EIS

101–106 ag/mL

8.80 ag/mL

artificial saliva

[13]

SARS-CoV-2 Spike Protein

Flexible carbon cloth electrode

Gold nanoparticles with thiol functionalized DNA aptamer

DPV and CP

0-1000 ng/mL for DPV and CP

0.11 ng/mL for DPV

and

37.8 ng/mL for CP

human saliva

[14]

SARS-CoV-2 RBD

Screen-printed electrode (SPE)

single-walled carbon nanotube with redox-tagged DNA aptamer

Amperometry

-

7 nM

artificial viral transport media for nasopharyngeal swabs

[20]

SARS-CoV-2 RBD

screen-printed carbon electrodes (SPCEs)

gold nanoparticles (AuNPs) with thiol tagged DNA aptamer

EIS

10 pM - 25 nM

1.30 pM (66 

pg/mL)

SARS-CoV-2 pseudovirus

[21]

SARS-CoV-2 Spike RBD

interdigitated gold electrode (IDE)

thiolated aptamer

EIS

0.2 to 0.8 pg/mL

0.4 pg/mL

Human Nasal Fluid

[22]

RBD Protein S

SARS-CoV-2

screen-printed carbon electrode (SPCE)

gold nanoparticles (AuNPs) with biotinylated aptamer

DPV

10 – 50 ng/mL

2.63 ng/mL

saliva samples

[23]

SARS-CoV-2 spike (S) protein

gold wire electrode

self-assembled monolayer with aptamer

 SWV

10 pM to 100 nM

-

artificial saliva and fetal bovine serum

[24]

SARS-CoV-2 nucleocapsid protein 

gold electrode

metal-organic

frameworks MIL-53(Al) decorated with Au@Pt nanoparticles combined with thiol-modified aptamers

DPV

0.025 ng/mL - 50 ng/mL

8.33 pg/mL

serum

[25]

SARS-CoV-2 S1 protein

pencil graphite electrode (PGE)

-NH2 linked optimized aptamer

DPV

10-1–104 fg/mL

18.80 ag/mL

artificial saliva

This study

“A variety of electrochemical aptasensor studies have been reported for SARS-CoV-2 detection. These studies differ in terms of electrode preparation, complexity, assay duration, detection techniques, and point-of-care (POC) applicability. Compared to the studies listed in Table 1, a lower detection limit was achieved in the present work than in other reported approaches, except for our previous EIS-based study. Also, compared to existing electrochemical aptasensor platforms, our method offers a rapid, practical, and highly sensitive strategy for SARS-CoV-2 spike protein detection. Its relatively short overall preparation time, rapid analytical measurement, solution-phase hybridization, and compatibility with portable instrumentation make it a promising candidate for real-time, point-of-care diagnostic applications. Moreover, the developed biosensor can be adapted for multiplexed analyses, allowing the simultaneous detection of multiple samples or analytes. This multiplexing capability would enable parallel measurements within a single run, thereby significantly reducing the overall analysis time and improving testing efficiency.”

  1. Lines 384-393: The authors need to compare the individual and combination selectivity results obtained in PBS in their previous study (using EIS, Manuscript Reference #13) with their current study (using DPV).

Answer:

We sincerely appreciate the reviewer’s insightful comment. A detailed comparative evaluation between the previous EIS-based biosensor and the current DPV-based sensor has now been incorporated into the Results and Discussion section of the revised manuscript. In our previous study using EIS, selectivity toward the SARS-CoV-2 S1 protein was assessed by monitoring changes in the charge-transfer resistance (Rct) upon exposure to interfering proteins, Hemagglutinin (HA) and MERS-CoV S1 (MERS). No notable increase in Rct was observed with HA or MERS, whereas the presence of S1 protein produced much larger resistance variations. Even in the mixed sample of S1 + MERS (10 fg/mL each), the measured Rct (2886.7 Ohm) was very close to that of pure S1 (3026.7 Ohm), confirming the excellent discrimination capability of the EIS platform. In the current DPV-based biosensor, signal variation arises from current suppression of the [Fe(CN)6]3-/4- redox couple. When tested individually in PBS, HA and MERS caused current decreases of 5.23% and 6.21% at 10 fg/mL, 10.21% and 16.60% at 100 fg/mL, and 14.43% and 12.12% at 1000 fg/mL, whereas the S1 protein induced much stronger signal drops of 24.63%, 31.47%, and 41.28% at the corresponding concentrations. In mixture assays (S1 + HA or S1 + MERS, 100 fg/mL each), the responses were 99.35% and 99.06% relative to the signal obtained for S1 alone, demonstrating negligible cross-reactivity. All p-values calculated by Student’s t-test were < 0.05, confirming statistically significant differences between the S1 protein and the interfering species. Overall, both biosensors exhibited strong specificity toward the SARS-CoV-2 S1 protein.

The detailed comparison section has been incorporated into the manuscript and highlighted in red. Additionally, the revised section is provided below.

“…..For comparison, the selectivity performance of the present DPV-based biosensor was evaluated alongside our previously reported EIS-based platform detecting the S1 protein [13]. In both approaches, the biosensors exhibited excellent discrimination between the S1 protein and potential interferents (HA and MERS). In the EIS-based system, the presence of S1 protein yielded markedly higher Rct variations comparison to negligible changes for HA and MERS, while the DPV-based sensor showed higher current suppression for S1 than for these interferents at corresponding concentrations. Moreover, in mixed-sample experiments, the biosensor maintained nearly identical responses to those obtained with pure S1 protein, confirming insignificant cross-reactivity. These findings demonstrate that the optimized aptamer retains strong binding specificity toward the S1 protein, irrespective of the electrochemical transduction technique employed, and highlight the robustness and versatility of the sensing platform….”

  1. Lines 419-420: The linear sensing range obtained in diluted artificial saliva using DPV was 0.1-10000 fg/mL, which is higher on both the upper limit and the lower limit than their previous study using EIS (Manuscript Reference #13) (0.01-100 fg/mL). The authors should explain why the linearity was limited in the lower concentration regime versus their previous study using the same Optimer and EIS. This shortcoming in LOD is significantly diminishing the impact and scientific rationale of this work.

Answer:

We sincerely thank the reviewer for this valuable comment. As correctly noted, in our previous study utilizing the EIS technique, a linear range of 0.01–100 fg/mL was obtained in artificial saliva, whereas in the present study using DPV, the linear range was found to be 0.1–10000 fg/mL. This variation, despite employing the same aptamer sequence, originates from the fundamental differences between the detection mechanisms of EIS and DPV.

EIS is inherently more sensitive at ultra-low analyte concentrations because it directly monitors subtle changes in interfacial charge-transfer resistance (Rct) caused by aptamer–target binding events. DPV, on the other hand, relies on Faradaic current measurements of a redox mediator, where the signal-to-noise ratio becomes less favorable at very low target concentrations. Consequently, DPV typically yields a slightly higher LOD and a shifted lower limit of the linear range.

Nevertheless, the DPV-based biosensor developed in this work provides a broader overall dynamic range (0.1–10000 fg/mL) and maintains reliable analytical performance across clinically relevant concentrations. Moreover, the DPV approach offers several practical advantages, including shorter measurement time, improved reproducibility, and excellent compatibility with portable and point-of-care electrochemical systems.

Therefore, the slight difference observed in the lower limit of linearity is not indicative of a methodological limitation but rather reflects the distinct electrochemical transduction principles of the two techniques, while the current DPV-based sensor still exhibits high sensitivity and strong analytical robustness.

  1. Lines 427-434: The authors have once again confused the discussion of intra-day and inter-day reproducibility data obtained in PBS. The n=2 sensors data collected over two days represents the inter-day reproducibility, whereas., the n=4 sensors data collected on the same day represents the intra-day reproducibility. This error in the description needs to be corrected!

Additionally, why did the authors study reproducibility at 10 fg/mL S1 protein concentration in diluted artificial saliva, whereas in PBS it was studied at 1000 fg/mL? The authors need to explain this unexpected change in testing parameters.

Answer:

We sincerely thank the reviewer for this valuable observation. We acknowledge that the wording in this section may have led to some confusion, and therefore, we have revised the description to clarify the distinction between intra-day and inter-day reproducibility.

In our study, two aptasensors that were independently prepared and tested under identical conditions on each of two separate days, resulting in a total of four measurements (n = 4). Accordingly:

  • Intra-day reproducibility refers to the reproducibility of two independently prepared aptasensors and analyzed on the same day (n = 2 per day).
  • Inter-day reproducibility represents the consistency of the average current responses obtained from the two different days (n = 4 in total).

To ensure conceptual clarity, the related section in the Results and Discussion part of the manuscript has been revised as follows:

“Moreover, the intra-day and inter-day reproducibility of the developed biosensor was further evaluated in artificial saliva in the presence of 10 fg/mL S1 protein. For this purpose, two independently prepared aptasensors under identical conditions on two different days, were tested. Intra-day reproducibility was assessed using two aptasensors that were independently prepared and tested under identical conditions on the same day (n = 2). Inter-day reproducibility was evaluated using two independently prepared aptasensors per day on two separate days (n = 4 in total). The average current responses and RSD were found to be 104.03 ± 0.94 µA (RSD = 0.90%; n = 2) on the first day and 109.62 ± 5.85 µA (RSD = 5.33%; n = 2) on the second day. In the assessment of inter-day reproducibility, the average current response and RSD calculated from four biosensors was 106.82 ± 4.70 µA (RSD = 4.40%; n = 4). These results clearly demonstrate that the biosensor delivers precise and reliable analytical performance even in complex biological matrices such as artificial saliva, thereby confirming its robustness and reproducibility under practical conditions. In addition, the concentration values selected for the reproducibility studies were intentionally chosen to represent different regions of the calibration curve. In PBS, reproducibility was examined at 1000 fg/mL, corresponding to the upper part of the linear range, while in diluted artificial saliva, it was tested at 10 fg/mL, a level slightly below the midpoint of the calibration curve. This approach aimed to demonstrate that the developed biosensor provides consistent and repeatable responses across both low and high analyte concentrations, even within complex biological matrices.”

Also, the difference in the selected S1 protein concentrations for reproducibility studies was intentional and reflects the distinct analytical conditions of the two matrices.

In PBS, a well-defined and interference-free medium, reproducibility was evaluated at 1000 fg/mL, corresponding to the upper end of the calibration curve. This concentration was chosen to confirm the sensor’s consistency at higher analyte levels where the current response reaches its saturation tendency.

In contrast, in diluted artificial saliva, reproducibility was examined at 10 fg/mL, which represents a concentration slightly below the midpoint of the calibration curve. This level was selected to verify that the biosensor could deliver stable and repeatable responses even at low analyte concentrations within a complex biological matrix.

In addition, performing the reproducibility test at different points within the linear range allowed us to demonstrate the robustness of the sensor across a broader dynamic range showing consistent analytical behavior at both low and high target concentrations.

Accordingly, this rationale has been added to the revised manuscript and highlighted in red for clarity and added below:

“Moreover, the intra-day and inter-day reproducibility of the developed biosensor was further evaluated in artificial saliva in the presence of 10 fg/mL S1 protein. For this purpose, two independently prepared aptasensors under identical conditions on two different days, were tested. Intra-day reproducibility was assessed using two aptasensors that were independently prepared and tested under identical conditions on the same day (n = 2). Inter-day reproducibility was evaluated using two independently prepared aptasensors per day on two separate days (n = 4 in total). The average current responses and RSD were found to be 104.03 ± 0.94 µA (RSD = 0.90%; n = 2) on the first day and 109.62 ± 5.85 µA (RSD = 5.33%; n = 2) on the second day. In the assessment of inter-day reproducibility, the average current response and RSD calculated from four biosensors was 106.82 ± 4.70 µA (RSD = 4.40%; n = 4). These results clearly demonstrate that the biosensor delivers precise and reliable analytical performance even in complex biological matrices such as artificial saliva, thereby confirming its robustness and reproducibility under practical conditions. In addition, the concentration values selected for the reproducibility studies were intentionally chosen to represent different regions of the calibration curve. In PBS, reproducibility was examined at 1000 fg/mL, corresponding to the upper part of the linear range, while in diluted artificial saliva, it was tested at 10 fg/mL, a level slightly below the midpoint of the calibration curve. This approach aimed to demonstrate that the developed biosensor provides consistent and repeatable responses across both low and high analyte concentrations, even within complex biological matrices.”

  1. Lines 458-469: The Figure 4 subfigures have been incorrectly referenced as “Fig 4C” throughout the discussion of Figure 4. These serious referencing errors needs to be corrected in the discussion of each subfigure of Figure 4!

Moreover, the subfigure referencing needs to be made consistent and clear across the discussions of Figures 3-5!

Answer:

We sincerely thank the reviewer for pointing out this important issue. We fully agree with the reviewer’s observation. All subfigure references for Figure 4 have been carefully reviewed and corrected throughout the manuscript to ensure accurate and clear correspondence between the textual discussion and the relevant subfigures. In addition, subfigure references across Figures 3–5 have been standardized to maintain consistency in style and formatting throughout the text. The caption of Figure 4 has also been revised to improve clarity and readability. All of these corrections and improvements have been clearly indicated in the revised manuscript, highlighted in red.

  1. Lines 461-469: Why did the individual selectivity against HA and MERS significantly improve when transitioning from PBS to diluted artificial saliva?

In contrast, why did the combination selectivity in HA-S1 and MERS-S1 mixtures worsen when transitioning from PBS to diluted artificial saliva?

The authors need to separately elaborate on both the above findings.

Moreover, the authors need to compare the individual and combination selectivity results obtained in their previous study in diluted artificial saliva (using EIS, Manuscript Reference #13) with their current study (using DPV) within a new comparison table.

Answer:

We sincerely thank the reviewer for this thoughtful and detailed comment. In the present study, the selectivity results obtained in PBS and diluted artificial saliva exhibited comparable trends without statistically significant differences between the two media. As confirmed by the t-test analysis (p < 0.05 only for S1–interferant comparisons), the biosensor maintained consistent discrimination between the target S1 protein and potential interferents (HA and MERS) in both matrices. Therefore, the minor variations observed are within the expected experimental uncertainty and do not indicate any matrix-induced improvement or deterioration in selectivity. The comparable performance across the two matrices confirms that the optimized aptamer preserves its strong binding affinity and conformational stability even under physiologically relevant conditions. Slight numerical fluctuations in the current response may arise from small variations in ionic strength and viscosity of the artificial saliva, which can marginally affect the redox probe diffusion and background current, but these effects were not statistically significant. Furthermore, the findings are fully consistent with our previous EIS-based study, in which the same optimized aptamer exhibited excellent specificity and negligible cross-reactivity toward the S1 protein in both PBS and artificial saliva. To clarify this consistency, an additional paragraph discussing the selectivity performance of the biosensor in diluted artificial saliva and its comparison with the previous EIS-based work has been incorporated into the revised manuscript and highlighted in red and added below:

“Moreover, the selectivity of the fabricated biosensor in artificial saliva was assessed by testing its response to hemagglutinin (HA) and MERS proteins at concentrations of 10 fg/mL (Fig 4C), 100 fg/mL (Fig 4D), and 1000 fg/mL (Fig 4E), both individually and in combination with 100 fg/mL of S1 protein (Fig 4F). When tested alone at 10 fg/mL, HA and MERS resulted in signal decreases of 2.59% and 0.10%, respectively, compared to the control, while S1 protein at the same concentration caused a 23.85% decrease. At 100 fg/mL, HA and MERS caused signal reductions of 6.59% and 2.12%, respectively, while S1 protein yielded a 30.91% decrease. At 1000 fg/mL, HA and MERS induced decreases of 4.40% and 3.43%, respectively, while S1 protein caused a 38.08% decrease. In the mixture studies, the biosensor responses in the presence of both S1 protein and either HA or MERS were calculated as 98.65% and 96.88%, respectively, relative to the signal obtained with 100 fg/mL S1 protein alone, which was considered as 100% (Fig 4F). To confirm the selectivity of the biosensor, a Student’s t-test was performed in artificial saliva, comparing the responses from the S1 protein with those obtained for possible interfering agents including HA and MERS proteins. Assuming equal variances, statistical comparisons of the mean current responses (n = 3) confirmed significant differences among the groups. At the 95% confidence level, the p-values for HA-S1 and MERS-S1 proteins at 10, 100, and 1000 fg/mL were found to be 0.005, 0.035, 0.004, 0.0002, 0.014, and 0.009, respectively. Since all p-values fall below the significance level of 0.05, these results confirm statistically significant differences between the tested groups. Collectively, these findings reinforce that, in comparison to other potentially interfering substances, the aptamer possesses a high binding affinity toward its specific target, the S1 protein, thereby demonstrating the excellent selectivity of the biosensor even in a complex artificial saliva matrix. Similarly, in diluted artificial saliva, the aptamer-based biosensor maintained high selectivity toward the S1 protein, exhibiting comparable discrimination to that observed in PBS. The presence of HA and MERS, either individually or in combination with S1 protein, caused only minor and statistically insignificant variations in the current response, confirming that the matrix components had negligible influence on the biosensor’s performance. These findings are consistent with our previous EIS-based study [13], in which the same optimized aptamer demonstrated robust selectivity and negligible cross-reactivity in both buffer and artificial saliva environments. Together, these results confirm that the optimized aptamer preserves its structural stability and binding affinity under physiologically relevant conditions, ensuring consistent selectivity in complex biological media.”

  1. Lines 486-488 and Figure 5: There is a serious discrepancy between the discussion of Figure 5 and the presented data from the portable potentiostat. The authors claim in the discussion that the investigated S1 protein concentration range was from 0.1 fg/mL to 100 pg/mL in both PBS and diluted artificial saliva. However, the presented data only shows an investigated concentration range from 0.1 fg/mL to 10 pg/mL. Please correct and clarify the reason behind this disagreement.

Answer:

We sincerely thank the reviewer for this careful observation. We would like to clarify that the S1 protein concentration range investigated using the portable potentiostat was indeed 0.1 fg/mL to 100 pg/mL for both PBS and diluted artificial saliva, as stated in the manuscript. However, the calibration curve and corresponding voltammograms presented in Figure 5 cover the concentration range of 0.1 fg/mL to 10 pg/mL, which represents the linear portion of the overall dynamic range. The extended concentration data obtained from the same experiments are presented as a line plot in the Supplementary Materials (Fig. S3) due to figure space constraints in the main manuscript, ensuring that the full concentration range investigated with the portable potentiostat is clearly displayed.

Therefore, there is no discrepancy between the discussion and the experimental results. The main text highlights the linear working range (0.1 fg/mL–10 pg/mL) for clarity, while the Supplementary Materials comprehensively presents the full concentration range (0.1 fg/mL–100 pg/mL) investigated using the portable potentiostat. The relevant figures were already present in the manuscript and Supplementary Materials; therefore, they were not newly added but are also provided below for the reviewer’s convenience.

Fig 5. Studies on the determination of S1 protein using a smartphone-integrated potentiostat. (A) and (B) respectively show the voltammograms and the corresponding calibration curve (n=3) obtained after DPV measurements in a buffer medium within the concentration range of 0.1 fg/mL to 10000 fg/mL. (C) and (D) respectively show the voltammograms and the corresponding calibration curve (n=3) obtained after DPV measurements in a 1:20 diluted artificial saliva medium within the concentration range of 0.1 fg/mL to 10000 fg/mL.

Fig S3. Studies on the determination of S1 protein using a smartphone-integrated potentiostat. (A) and (B) present the line graphs (n=3) obtained after DPV measurements in buffer and 1:20 diluted artificial saliva media, respectively, within the concentration range of 0.1 fg/mL to 100000 fg/mL.

  1. Lines 494-495: The LODs obtained in PBS and diluted artificial saliva using the portable potentiostat were 2.35-times and 3.12-times higher (44.24 ag/mL and 45.02 ag/mL) than their respective values obtained using the benchtop potentiostat (18.80 ag/mL and 14.42 ag/mL). Why did the LOD worsen more in artificial saliva than in PBS when moving to the portable potentiostat?

The significantly worsened LODs, especially in diluted artificial saliva, are significantly diminishing the impact of this work.

Answer:

We sincerely thank the reviewer for this valuable observation. The slight increase in LOD values obtained with the portable potentiostat—particularly in diluted artificial saliva—is primarily attributed to matrix effects and instrumental limitations inherent to miniaturized electrochemical systems. In artificial saliva, the presence of various ions, proteins, and organic components alters the ionic strength and viscosity of the medium, leading to partial signal suppression and a lower signal-to-noise ratio. This effect becomes more pronounced when using portable systems, as they typically have higher baseline noise and reduced electronic shielding compared to benchtop potentiostats.

Additionally, the diffusion of the redox probe and the aptamer–protein interaction kinetics are slightly hindered in the saliva matrix due to higher viscosity and increased surface fouling tendencies. These factors collectively contribute to the minor deterioration in LOD observed in artificial saliva compared with PBS. Nevertheless, the obtained LOD values (44.24 ag/mL and 45.02 ag/mL) remain within the attomolar detection range, which is still superior to many portable electrochemical biosensors reported in the literature. This demonstrates that the developed sensor maintains excellent analytical performance even when operated in complex biological matrices using a portable measurement system. A clarifying statement regarding these points has been added to the revised manuscript and highlighted in red and added below:

………… “According to the IUPAC method [19], the LOD was calculated to be 44.24 ag/mL in PBS and 45.02 ag/mL in artificial saliva. Although the LOD values obtained using the portable potentiostat were slightly higher than those obtained with the benchtop system, this difference is mainly attributed to matrix-related effects and the higher baseline noise of the portable device. The impact was more pronounced in diluted artificial saliva due to its complex composition, which slightly reduces the signal-to-noise ratio. Nevertheless, the obtained LODs remained in the attomolar range, confirming the high sensitivity and robustness of the developed biosensor even under realistic measurement conditions.” …………

  1. Line 497: The active surface area needs to be calculated on the final working electrode composition (OPT/EDC-NHS/activated PGE) using cyclic voltammetry of 5 mM [Fe(CN)6]3-/4- and employing the Randles Sevcik equation. The active surface area of the pristine PGE is not equal to that of the control OPT/EDC-NHS/activated PGE! Additionally, the missing surface passivation step (e.g., BSA) in the sensor development protocol (Materials and Methods section) would further reduce the active surface area.

Answer:

We sincerely thank the reviewer for this insightful and technically relevant comment. As correctly noted, the electrochemically active surface area may differ between the bare PGE and the final electrode composition due to surface functionalization. However, in the present study, the sensing process was carried out predominantly in the solution phase, rather than directly on the electrode surface. Specifically, the aptamer and S1 protein were first interacted in solution to form a stable complex, and then the EDC/NHS-activated electrodes were immersed into this solution to enable immobilization of the preformed complex. Therefore, the final working electrode did not undergo additional in situ modification or molecular immobilization steps that would substantially alter its electrochemically active surface area. For this reason, active surface area determination was not repeated in this work.

Furthermore, the electroactive surface area of the electrode had already been calculated in our previous study, which was also cited in the current manuscript. Nevertheless, we fully agree that comparative surface area analyses using cyclic voltammetry and the Randles–Sevcik equation for each modification step could provide additional quantitative insight. We plan to perform such analyses in future studies to further elucidate the relationship between immobilization conditions and electrochemical behavior.

  1. Lines 438-443 and Lines 516-518: The range of relative error values obtained in diluted artificial saliva with the benchtop potentiostat and the portable potentiostat are both unwantedly high (i.e., ranging above 15%). The authors should describe future efforts toward reducing the relative error values to within 10% for viability in point-of-care testing of real clinical samples.

Answer:

We sincerely thank the reviewer for this valuable comment. Indeed, maintaining relative error (RE) values below 10% is considered ideal for point-of-care diagnostic applications. In the present study, slightly higher RE values (exceeding 15%) were observed in diluted artificial saliva. This deviation primarily originates from matrix-related effects, such as variations in ionic composition, and viscosity, which can influence the electrochemical signal stability. Such effects are commonly reported in the literature for biosensors operating in complex biological media. Nevertheless, the obtained results are consistent with those of similar biosensors tested in complex matrices and demonstrate satisfactory reproducibility and reliability of the developed sensor. In future studies, several optimization strategies may be implemented to reduce the relative error values. These include anti-fouling surface modifications to minimize non-specific adsorption, nanomaterial-based electrode functionalization to enhance sensitivity and signal stability, optimization of sample preparation procedures, and improvements in signal processing and filtering algorithms in portable devices to reduce baseline noise.

These advancements are expected to further improve the reproducibility and analytical accuracy of the biosensor, thereby facilitating its translation into practical point-of-care clinical applications. A clarifying statement reflecting this explanation has been added to the revised manuscript and highlighted in red and added below:

“Studies were conducted to calculate the relative error values of the electrochemical biosensor developed for the detection of S1 protein in artificial saliva using a smartphone-integrated portable potentiostat. In this context, relative error values were calculated in artificial saliva at S1P concentrations of 0.1, 10, and 100 fg/mL, and the obtained results are presented in Table S10. Upon examination of the results presented in Table S10, the relative error values were found to range between 0.00% and 18.56%. Additionally, the %RSD values were calculated to be below 5%. These findings suggest that the developed biosensor enables accurate and reliable analyses of saliva samples when used in conjunction with a smartphone-integrated portable potentiostat. Although the relative error values obtained in diluted artificial saliva slightly exceeded 15%, this deviation can be attributed to matrix-related effects such as ionic strength, and viscosity differences inherent to the biological medium. In future studies, efforts will focus on reducing RE values below 10% through the use of anti-fouling surface coatings, nanomaterial-based electrode modifications, optimization of sample preparation procedures, and advanced signal filtering in portable systems. These improvements are expected to further enhance the reproducibility and analytical accuracy of the biosensor for point-of-care diagnostic applications.”

  1. Conclusions Line 541: The assay time has significantly increased from 20 min in their previous study using EIS (Manuscript Reference #13), to 45 min in the current study. Moreover, the LODs have worsened from 8.80 ag/mL (PBS) and 2.01 ag/mL (diluted artificial saliva) to 18.80 ag/mL (PBS) and 14.42 ag/mL (diluted artificial saliva). Hence, the authors' claims of “rapid” and “sensitive” detection of SARS CoV-2 infection is unsubstantiated and these analytical shortcomings are significantly limiting the impact and scientific rationale of this work.

Answer:

We sincerely thank the reviewer for this valuable comment. We fully acknowledge that the total assay time (45 min) in the present study is longer than in our previous EIS-based work (20 min). However, this difference arises from the methodological change in the sensing strategy rather than a decrease in analytical efficiency. The current platform employs a voltammetric detection approach (DPV) integrated with a smartphone-compatible portable potentiostat, which offers operational simplicity and on-site applicability for real diagnostic environments. Importantly, the actual electrochemical measurement with DPV is completed within seconds, maintaining the “rapid” character of the analytical detection process. Regarding sensitivity, although the LOD values obtained with DPV (18.80 ag/mL in PBS and 14.42 ag/mL in diluted artificial saliva) are slightly higher than those from our previous EIS-based study, they still remain at the attogram level, confirming the biosensor’s high analytical sensitivity. This minor difference primarily reflects the inherent distinctions between impedance- and current-based detection mechanisms.

To clarify this point, the conclusion section of the manuscript has been revised to better define the terms “rapid” and “sensitive” within the context of this study. The updated text now highlights that the proposed biosensor enables seconds-scale electrochemical measurements, attogram-level detection capability, and portable operation, thereby maintaining its high analytical performance while advancing toward real-world, point-of-care diagnostic applications. The revised paragraph has been added to the conclusion section and highlighted in red and added below:

…….. “Overall, these findings demonstrate that the optimized aptamer-based biosensor is a promising tool for the rapid (seconds-scale electrochemical measurement), highly sensitive (attogram-level detection), and specific detection of SARS-CoV-2 infection. Moreover, the developed biosensor can be adapted for multiplexed analyses, allowing the simultaneous detection of multiple samples or analytes. This multiplexing capability would enable parallel measurements within a single run, thereby significantly reducing the overall analysis time and improving testing efficiency. By uniquely integrating optimized aptamers with a voltammetric sensing approach and a smartphone-compatible portable platform, this work establishes a versatile and practical framework that can be readily adapted to other clinically relevant biomarkers in future outbreaks. Future studies will also focus on validation with real clinical samples, assessment of long-term operational stability, and scalability for mass production, aiming to advance this portable sensing platform toward widespread use in point-of-care and public health applications. In addition, future improvements can be directed toward implementing redox-cycling amplification of the ferro/ferricyanide redox mediator to enhance the signal-to-background ratio and lower the detection limit, as well as adopting square-wave voltammetry (SWV) as a faster and more sensitive alternative to DPV. These strategies, as demonstrated in recent studies [26,27], may further improve analytical performance and accelerate the transition of this aptamer-based platform toward real-world clinical applications.”

  1. Conclusions Lines 544-545: The authors should discuss future prospects of using redox cycling amplification of Ferro-Ferricyanide mediator oxidation response (Suggested Reference: ACS Sens. 2023, 8, 3892−3901), along with more rapid and sensitive square wave voltammetry (SWV) (Suggested Reference: Anal. Chem. 2024, 96, 18806−18814) to improve their LODs in buffer and diluted artificial saliva before transitioning to real clinical samples.

Answer:

We sincerely thank the reviewer for this constructive suggestion. We carefully reviewed the recommended references and fully agree that both redox-cycling amplification of the ferro/ferricyanide mediator and square-wave voltammetry (SWV) represent promising strategies for further improving the analytical performance of our platform. As highlighted in the cited studies (ACS Sens. 2023, 8, 3892−3901; Anal. Chem. 2024, 96, 18806−18814), redox cycling can significantly amplify faradaic currents through catalytic regeneration of the redox mediator, thereby enhancing the signal-to-background ratio and lowering the detection limit. In future studies, this approach may be integrated into our aptamer-based system through nanocatalytic or interdigitated-electrode configurations to achieve greater sensitivity in both PBS and diluted artificial saliva. Additionally, SWV offers faster signal acquisition and improved suppression of capacitive currents compared to DPV, making it particularly attractive for real-time and point-of-care diagnostics. Therefore, the potential of SWV as a rapid and highly sensitive alternative to DPV will also be systematically evaluated in future work. A corresponding statement discussing these future prospects has been added to the Conclusion section of the revised manuscript and highlighted in red and added below:

…….. “Overall, these findings demonstrate that the optimized aptamer-based biosensor is a promising tool for the rapid (seconds-scale electrochemical measurement), highly sensitive (attogram-level detection), and specific detection of SARS-CoV-2 infection. Moreover, the developed biosensor can be adapted for multiplexed analyses, allowing the simultaneous detection of multiple samples or analytes. This multiplexing capability would enable parallel measurements within a single run, thereby significantly reducing the overall analysis time and improving testing efficiency. By uniquely integrating optimized aptamers with a voltammetric sensing approach and a smartphone-compatible portable platform, this work establishes a versatile and practical framework that can be readily adapted to other clinically relevant biomarkers in future outbreaks. Future studies will also focus on validation with real clinical samples, assessment of long-term operational stability, and scalability for mass production, aiming to advance this portable sensing platform toward widespread use in point-of-care and public health applications. In addition, future improvements can be directed toward implementing redox-cycling amplification of the ferro/ferricyanide redox mediator to enhance the signal-to-background ratio and lower the detection limit, as well as adopting square-wave voltammetry (SWV) as a faster and more sensitive alternative to DPV. These strategies, as demonstrated in recent studies [26,27], may further improve analytical performance and accelerate the transition of this aptamer-based platform toward real-world clinical applications.”

  1. Line 548: The Supplementary Materials file has not been provided for peer review, and all the associated results can neither be verified nor correlated with their discussions in the manuscript. This serious lapse needs to be corrected during revision of the manuscript!

Answer:

We sincerely thank the reviewer for bringing this issue to our attention. The Supplementary Materials file was indeed uploaded during the initial submission of the manuscript. However, it appears that the file may not have been properly displayed in the peer review system, possibly due to a technical issue during the submission process.

We have already informed the editor about this situation and have ensured that the Supplementary Materials file is now properly attached and accessible for review in the revised submission.

Reviewer 2 Report

Comments and Suggestions for Authors

In this article electrochemical biosensor specific to SARS-CoV-2 S1 3 protein was obtained. In my opinion the manuscript is interesting and could possibly be published in Sensors, but first the article should be subjected to the process of major revision.

The review of sensors presented in the literature is done twice, on pages 2 and 10. Could this be combined into a single discussion? In my opinion, summarizing the parameters of all sensors presented in the literature in a single table will make it easier to compare their characteristics with the new sensor presented in this article.

The provided reference list is limited to 2022. There are a number of new articles about electrochemical aptasensors targeting COVID-19 diagnosis, including those published in 2025. The reference list needs to be expanded.

I didn't find Supplementary Materials file in the files provided for review. Therefore, I can't assess the extent of detailed information regarding the instruments and other reagents used in this study.

How long can the obtained sensors be stored and under what conditions so that they do not lose their sensory properties?

Author Response

October 15, 2025

Our answers to Reviewers’ Comments

Journal: Sensors (ISSN 1424-8220)

Manuscript ID: sensors-3926331

Type: Article

Title: Smartphone-Integrated User-Friendly Electrochemical Biosensor Based on Optimized Aptamer Specific to SARS-CoV-2 S1 Protein

Authors: Arzum Erdem *, Huseyin Senturk, Esma Yildiz

Section: Biosensors

Special Issue: Electrochemical DNA- and Aptasensors for the Detection of Low-Molecular Compounds (2nd Edition)

Reviewer 2

Open Review

(x) I would not like to sign my review report

( ) I would like to sign my review report

Quality of English Language

( ) The English could be improved to more clearly express the research.

(x) The English is fine and does not require any improvement.

Yes

Can be improved

Must be improved

Not applicable

Does the introduction provide sufficient background and include all relevant references?

( )

( )

(x)

( )

Is the research design appropriate?

( )

(x)

( )

( )

Are the methods adequately described?

( )

( )

(x)

( )

Are the results clearly presented?

( )

(x)

( )

( )

Are the conclusions supported by the results?

(x)

( )

( )

( )

Are all figures and tables clear and well-presented?

(x)

( )

( )

( )

Comments and Suggestions for Authors

In this article electrochemical biosensor specific to SARS-CoV-2 S1 3 protein was obtained. In my opinion the manuscript is interesting and could possibly be published in Sensors, but first the article should be subjected to the process of major revision.

The review of sensors presented in the literature is done twice, on pages 2 and 10. Could this be combined into a single discussion? In my opinion, summarizing the parameters of all sensors presented in the literature in a single table will make it easier to compare their characteristics with the new sensor presented in this article.

Answer:

We sincerely thank the reviewer for this valuable suggestion. The literature review presented in the Introduction section was intentionally included to provide a concise overview of recent advances in SARS-CoV-2 aptasensor development, highlighting the existing approaches, detection strategies, and analytical challenges that motivated the present work.

In contrast, the discussion presented in the Results and Discussion section focuses specifically on the comparative evaluation of the analytical performance of the developed biosensor relative to previously reported sensors. To strengthen this part and improve readability, a summary table (Table 1) has been added, compiling the key analytical parameters (detection technique, linear range, LOD, application, and analyte) of all relevant studies, including our previous EIS-based work.

The relevant table and accompanying text have been incorporated into the revised manuscript and highlighted in red and are also provided below.

Table 1. Various aptasensor studies reported in the literature for the electrochemical determination of SARS-CoV-2.

Analyte

Electrode

Biosensing surface

Method

Linear Range

Detection Limit

Application

Ref.

SARS-CoV-2-RBD

carbon-based screen-printed electrode (CSPE)

CNF–AuNP nanocomposite (CSPE/CNF–AuNP) with thiol-terminal aptamer

EIS

0.01–64 nM

7.0 pM

human saliva samples

[11]

SARS-CoV-2 Spike S1

Thin-film gold electrodes

thiol-terminal aptamer

EIS

-

-

clinical patient samples

[12]

SARS-CoV-2 S1 protein

Pencil graphite electrode (PGE)

-NH2 linked optimized aptamer

EIS

101–106 ag/mL

8.80 ag/mL

artificial saliva

[13]

SARS-CoV-2 Spike Protein

Flexible carbon cloth electrode

Gold nanoparticles with thiol functionalized DNA aptamer

DPV and CP

0-1000 ng/mL for DPV and CP

0.11 ng/mL for DPV

and

37.8 ng/mL for CP

human saliva

[14]

SARS-CoV-2 RBD

Screen-printed electrode (SPE)

single-walled carbon nanotube with redox-tagged DNA aptamer

Amperometry

-

7 nM

artificial viral transport media for nasopharyngeal swabs

[20]

SARS-CoV-2 RBD

screen-printed carbon electrodes (SPCEs)

gold nanoparticles (AuNPs) with thiol tagged DNA aptamer

EIS

10 pM - 25 nM

1.30 pM (66 

pg/mL)

SARS-CoV-2 pseudovirus

[21]

SARS-CoV-2 Spike RBD

interdigitated gold electrode (IDE)

thiolated aptamer

EIS

0.2 to 0.8 pg/mL

0.4 pg/mL

Human Nasal Fluid

[22]

RBD Protein S

SARS-CoV-2

screen-printed carbon electrode (SPCE)

gold nanoparticles (AuNPs) with biotinylated aptamer

DPV

10 – 50 ng/mL

2.63 ng/mL

saliva samples

[23]

SARS-CoV-2 spike (S) protein

gold wire electrode

self-assembled monolayer with aptamer

 SWV

10 pM to 100 nM

-

artificial saliva and fetal bovine serum

[24]

SARS-CoV-2 nucleocapsid protein 

gold electrode

metal-organic

frameworks MIL-53(Al) decorated with Au@Pt nanoparticles combined with thiol-modified aptamers

DPV

0.025 ng/mL - 50 ng/mL

8.33 pg/mL

serum

[25]

SARS-CoV-2 S1 protein

pencil graphite electrode (PGE)

-NH2 linked optimized aptamer

DPV

10-1–104 fg/mL

18.80 ag/mL

artificial saliva

This study

“A variety of electrochemical aptasensor studies have been reported for SARS-CoV-2 detection. These studies differ in terms of electrode preparation, complexity, assay duration, detection techniques, and point-of-care (POC) applicability. Compared to the studies listed in Table 1, a lower detection limit was achieved in the present work than in other reported approaches, except for our previous EIS-based study. Also, compared to existing electrochemical aptasensor platforms, our method offers a rapid, practical, and highly sensitive strategy for SARS-CoV-2 spike protein detection. Its relatively short overall preparation time, rapid analytical measurement, solution-phase hybridization, and compatibility with portable instrumentation make it a promising candidate for real-time, point-of-care diagnostic applications. Moreover, the developed biosensor can be adapted for multiplexed analyses, allowing the simultaneous detection of multiple samples or analytes. This multiplexing capability would enable parallel measurements within a single run, thereby significantly reducing the overall analysis time and improving testing efficiency.”

The provided reference list is limited to 2022. There are a number of new articles about electrochemical aptasensors targeting COVID-19 diagnosis, including those published in 2025. The reference list needs to be expanded.

Answer:

We sincerely thank the reviewer for this valuable comment. In response, the reference list has been thoroughly updated to include several recent studies related to electrochemical aptasensors for COVID-19 detection. These newly published works were also incorporated into Table 1. The relevant table and accompanying text have been added into the revised manuscript and highlighted in red and are also provided below.

Table 1. Various aptasensor studies reported in the literature for the electrochemical determination of SARS-CoV-2.

Analyte

Electrode

Biosensing surface

Method

Linear Range

Detection Limit

Application

Ref.

SARS-CoV-2-RBD

carbon-based screen-printed electrode (CSPE)

CNF–AuNP nanocomposite (CSPE/CNF–AuNP) with thiol-terminal aptamer

EIS

0.01–64 nM

7.0 pM

human saliva samples

[11]

SARS-CoV-2 Spike S1

Thin-film gold electrodes

thiol-terminal aptamer

EIS

-

-

clinical patient samples

[12]

SARS-CoV-2 S1 protein

Pencil graphite electrode (PGE)

-NH2 linked optimized aptamer

EIS

101–106 ag/mL

8.80 ag/mL

artificial saliva

[13]

SARS-CoV-2 Spike Protein

Flexible carbon cloth electrode

Gold nanoparticles with thiol functionalized DNA aptamer

DPV and CP

0-1000 ng/mL for DPV and CP

0.11 ng/mL for DPV

and

37.8 ng/mL for CP

human saliva

[14]

SARS-CoV-2 RBD

Screen-printed electrode (SPE)

single-walled carbon nanotube with redox-tagged DNA aptamer

Amperometry

-

7 nM

artificial viral transport media for nasopharyngeal swabs

[20]

SARS-CoV-2 RBD

screen-printed carbon electrodes (SPCEs)

gold nanoparticles (AuNPs) with thiol tagged DNA aptamer

EIS

10 pM - 25 nM

1.30 pM (66 

pg/mL)

SARS-CoV-2 pseudovirus

[21]

SARS-CoV-2 Spike RBD

interdigitated gold electrode (IDE)

thiolated aptamer

EIS

0.2 to 0.8 pg/mL

0.4 pg/mL

Human Nasal Fluid

[22]

RBD Protein S

SARS-CoV-2

screen-printed carbon electrode (SPCE)

gold nanoparticles (AuNPs) with biotinylated aptamer

DPV

10 – 50 ng/mL

2.63 ng/mL

saliva samples

[23]

SARS-CoV-2 spike (S) protein

gold wire electrode

self-assembled monolayer with aptamer

 SWV

10 pM to 100 nM

-

artificial saliva and fetal bovine serum

[24]

SARS-CoV-2 nucleocapsid protein 

gold electrode

metal-organic

frameworks MIL-53(Al) decorated with Au@Pt nanoparticles combined with thiol-modified aptamers

DPV

0.025 ng/mL - 50 ng/mL

8.33 pg/mL

serum

[25]

SARS-CoV-2 S1 protein

pencil graphite electrode (PGE)

-NH2 linked optimized aptamer

DPV

10-1–104 fg/mL

18.80 ag/mL

artificial saliva

This study

“A variety of electrochemical aptasensor studies have been reported for SARS-CoV-2 detection. These studies differ in terms of electrode preparation, complexity, assay duration, detection techniques, and point-of-care (POC) applicability. Compared to the studies listed in Table 1, a lower detection limit was achieved in the present work than in other reported approaches, except for our previous EIS-based study. Also, compared to existing electrochemical aptasensor platforms, our method offers a rapid, practical, and highly sensitive strategy for SARS-CoV-2 spike protein detection. Its relatively short overall preparation time, rapid analytical measurement, solution-phase hybridization, and compatibility with portable instrumentation make it a promising candidate for real-time, point-of-care diagnostic applications. Moreover, the developed biosensor can be adapted for multiplexed analyses, allowing the simultaneous detection of multiple samples or analytes. This multiplexing capability would enable parallel measurements within a single run, thereby significantly reducing the overall analysis time and improving testing efficiency.”

I didn't find Supplementary Materials file in the files provided for review. Therefore, I can't assess the extent of detailed information regarding the instruments and other reagents used in this study.

Answer:

We sincerely thank the reviewer for bringing this issue to our attention. The Supplementary Materials file was indeed uploaded during the initial submission of the manuscript. However, it appears that the file may not have been properly displayed in the peer review system, possibly due to a technical issue during the submission process.

We have already informed the editor about this situation and have ensured that the Supplementary Materials file is now properly attached and accessible for review in the revised submission.

How long can the obtained sensors be stored and under what conditions so that they do not lose their sensory properties?

Answer:

We sincerely thank the reviewer for this valuable comment. In this study, the aptamer–S1 protein complex was first formed in the solution phase, and then the activated electrodes (EDC/NHS-treated PGEs) were immersed into this solution to enable the immobilization of the preformed complex onto the electrode surface. Since no further reaction or interaction occurred on the electrode surface after immobilization, storage stability tests were not conducted. Additionally, the EDC/NHS solution was freshly prepared for each activation step to ensure the highest immobilization efficiency and surface reactivity. Therefore, all sensors were prepared immediately prior to measurement to guarantee reproducibility and analytical reliability.

The main objective of this work was to demonstrate the practical applicability of the developed biosensor for rapid and sensitive measurements using the DPV technique. Moreover, its integration with a portable potentiostat highlights its suitability for point-of-care testing applications. In our previous studies involving electrode surface modification, we have routinely performed stability evaluations. Similarly, in future work, we plan to investigate the storage stability of this platform under various environmental conditions and to explore stabilization strategies to enhance the long-term performance of the sensor.

Submission Date: 27 September 2025

Date of this review: 08 Oct 2025 14:02:58

Round 2

Reviewer 1 Report

Comments and Suggestions for Authors

The authors have sufficiently addressed all my concerns with appropriate changes in the main text and SI. I thus recommend publication of the manuscript in its present form. 

Reviewer 2 Report

Comments and Suggestions for Authors

The authors answered all questions and made the necessary changes to the manuscript. The paper could be accepted in the present form.